# Predictors of mortality in patients with drug-resistant tuberculosis: A systematic review and meta-analysis

Ayinalem Alemu[1]*, Zebenay Workneh Bitew[2], Teshager Worku[3], Dinka Fikadu Gamtesa[1], Animut Alebel[4,5]

1 Ethiopian Public Health Institute, Addis Ababa, Ethiopia, 2 St Paul's Hospital Millennium Medical CollegeAddis Ababa, Addis Ababa, Ethiopia, 3 College of Health and Medical Sciences, Harar, Ethiopia, 4 College of Health Science, Debre Markos University, Debre Markos, Ethiopia, 5 School of Public Health, Faculty of Health, University of Technology Sydney, Sydney, Australia

* ayinalemal@gmail.com

## Abstract

### Background

Even though the lives of millions have been saved in the past decades, the mortality rate in patients with drug-resistant tuberculosis is still high. Different factors are associated with this mortality. However, there is no comprehensive global report addressing these risk factors. This study aimed to determine the predictors of mortality using data generated at the global level.

### Methods

We systematically searched five electronic major databases (PubMed/Medline, CINAHL, EMBASE, Scopus, Web of Science), and other sources (Google Scholar, Google). We used the Joanna Briggs Institute Critical Appraisal tools to assess the quality of included articles. Heterogeneity assessment was conducted using the forest plot and $I^2$ heterogeneity test. Data were analyzed using STATA Version 15. The pooled hazard ratio, risk ratio, and odd's ratio were estimated along with their 95% CIs.

### Result

After reviewing 640 articles, 49 studies met the inclusion criteria and were included in the final analysis. The predictors of mortality were; being male (HR = 1.25,95%CI;1.08,1.41, $I^2$;30.5%), older age (HR = 2.13, 95%CI;1.64,2.62,$I^2$;59.0%,RR = 1.40,95%CI; 1.26, 1.53, $I^2$; 48.4%) including a 1 year increase in age (HR = 1.01, 95%CI;1.00,1.03,$I^2$;73.0%), under-nutrition (HR = 1.62,95%CI;1.28,1.97,$I^2$;87.2%, RR = 3.13, 95% CI; 2.17,4.09, $I^2$;0.0%), presence of any type of co-morbidity (HR = 1.92,95%CI;1.50–2.33,$I^2$;61.4%, RR = 1.61, 95%CI;1.29, 1.93,$I^2$;0.0%), having diabetes (HR = 1.74, 95%CI; 1.24,2.24, $I^2$;37.3%, RR = 1.60, 95%CI;1.13,2.07, $I^2$;0.0%), HIV co-infection (HR = 2.15, 95%CI;1.69,2.61, $I^2$; 48.2%, RR = 1.49, 95%CI;1.27,1.72, $I^2$;19.5%), TB history (HR = 1.30,95%CI;1.06,1.54, $I^2$;64.6%), previous second-line anti-TB treatment (HR = 2.52, 95% CI;2.15,2.88, $I^2$;0.0%), being

**Data Availability Statement:** All relevant data are within the paper and its Supporting information files.

**Funding:** The author(s) received no specific funding for this work.

**Competing interests:** The authors have declared that no competing interests exist.

smear positive at the baseline (HR = 1.45, 95%CI;1.14,1.76, $I^2$;49.2%, RR = 1.58,95% CI;1.46,1.69, $I^2$;48.7%), having XDR-TB (HR = 2.01, 95%CI;1.50,2.52, $I^2$;60.8%, RR = 2.44, 95%CI;2.16,2.73,$I^2$;46.1%), and any type of clinical complication (HR = 2.98, 95%CI; 2.32, 3.64, $I^2$; 69.9%). There are differences and overlaps of predictors of mortality across different drug-resistance categories. The common predictors of mortality among different drug-resistance categories include; older age, presence of any type of co-morbidity, and undernutrition.

## Conclusion

Different patient-related demographic (male sex, older age), and clinical factors (undernutrition, HIV co-infection, co-morbidity, diabetes, clinical complications, TB history, previous second-line anti-TB treatment, smear-positive TB, and XDR-TB) were the predictors of mortality in patients with drug-resistant tuberculosis. The findings would be an important input to the global community to take important measures.

## Introduction

Tuberculosis (TB) is the top cause of mortality from a single infectious disease [1]. In addition to the low detection rate, poor treatment outcome is becoming a major challenge of TB [2]. The World Health Organization (WHO) identified and introduced a directly observed treatment, short-course (DOTS) strategy to improve the treatment cure rate of TB [3, 4]. The treatment usually takes six to eight months: however, it takes a longer time if drug-resistant tuberculosis (DR-TB) is diagnosed [4]. Drug-resistant tuberculosis is caused by *Mycobacterium* bacteria that are resistant to at least one first-line anti-TB drug [5]. Nowadays, the emergence of DR-TB has become a major public health challenge globally, notably in resources limited settings, and it is commonly associated with unsuccessful treatment outcomes [6]. When the bacteria become resistant to more anti-TB drugs such as MDR-TB and XDR-TB, the treatment outcome worsens [7]. According to the 2019 WHO estimate, the global treatment success rate of MDR/RR-TB was 56% and XDR-TB was 39% [1].

A high mortality rate was observed among patients with DR-TB globally. Different patient and programmatic related factors are contributing to this high mortality rate [8–13]. Patient-related determinants include demographic characteristics (age and sex), behavioral factors (smoking, alcohol use, and substance addiction), and clinical factors (comorbidities, *HIV*, undernutrition, anemia, clinical complications, adverse effects, and type of drug resistance). Programmatic management of drug-resistant TB is important to limit TB, prevent the emergence of DR-TB, and have a successful treatment outcome [5]. Though the mortality rate among DR-TB patients is high, it highly varies across countries and settings. Different predictors contribute to this unacceptable high level of mortality. Even though there are previously conducted systematic reviews regarding the poor treatment outcome of DR-TB and its predictors, most of the studies are geographically restricted or restricted to a certain study group [14–16]. For example, our team performed a systematic review and meta-analysis to assess the poor treatment outcome and its predictors among DR-TB patients in Ethiopia. The study estimate revealed that the proportion and incidence density rate of mortality among DR-TB patients in Ethiopia was 15.13% and 9.28/1000 person-months respectively. Besides, the study revealed that the predictors of poor treatment outcome include; older age undernutrition,

clinical complications, lower body weight, HIV positivity, anemia, non-HIV comorbidities, treatment delay, and extrapulmonary involvement [14]. However, there is limited information that specifically addressed the predictors of mortality among DR-TB patients at the global level. Thus, our systematic review and meta-analysis study aimed to assess the predictors of mortality among patients with DR-TB based on available studies globally.

## Methods

### Search strategy and study selection

This systematic review and meta-analysis was reported according to the Preferred Reporting Items for Systematic Reviews and Meta-Analyses (PRISMA) checklist [17, 18] (S1 Table). We systematically searched five major databases; PubMed/Medline, CINAHL, EMBASE, Scopus, and Web of Science. We also searched Google Scholar and Google for gray literature. The search was conducted from the 5th to the 20th of June 2020. We used the following keywords: Predictors, Indicators, Mortality, Drug-resistant, Tuberculosis. The keywords were searched in combination with the Boolean words AND/OR. The search string applied for the Ovid Embase database was ('predictors'/exp OR predictors OR 'indicators'/exp OR indicators) AND ('mortality'/exp OR mortality) AND 'drug resistant' AND (tuberculosis'/exp OR tuberculosis). Two authors (AA[1], TW) independently searched articles published in English under the guidance of a senior librarian working at the Ethiopian Public Health Institute and Haramaya University College of Health Science without the time and boundary restrictions (S2 Table). Original studies assessing the predictors of mortality in patients with DR-TB during anti-TB treatment were included. Drug-resistant tuberculosis, defined as when someone is infected with *Mycobacterium tuberculosis*, which is resistant to at least one first-line anti-TB drug. The laboratory diagnostic methods to rule out DR-TB could be conventional phenotypic drug-susceptibility tests or molecular methods like Xpert MTB/RIF assay and Line Probe Assay (MTBDR*plus*, MTBDR*sl*). Excluded were case reports and studies that included a mixed population (both DR-TB and drug-susceptible TB) (Fig 1) (S3 Table).

Based on the study questions and inclusion criteria, in the first stage, we screened articles for titles and abstracts. In the second stage, articles were assessed for full-text review. Two authors (ZWB and AA[1]) independently performed the study eligibility assessment. The inconsistencies were resolved through discussion, and PICOS (participants, interventions, comparison, outcome, and study setting) criteria were used to review the articles. Data were extracted from the included articles by two authors (AA[1] and ZWB). The exacted data were; author, publication year, study period, study population, country, study setting, study design, sample size, number of deaths, and total follow-up period (Table 1). Also, we extracted data for different predictors of mortality using crude HR, RR, and OR along with the 95% CI (Table 2). The extracted data were stored in Microsoft Excel 2016.

### PICOS criteria

Participants: Patients with drug-resistant tuberculosis.

Interventions: Anti-tuberculosis treatment.

Comparators: Alive in the treatment period.

Outcomes: Death from any cause during the treatment period among DR-TB patients.

Study type: Cohort and case-control studies.

Study setting: Any country in the globe.

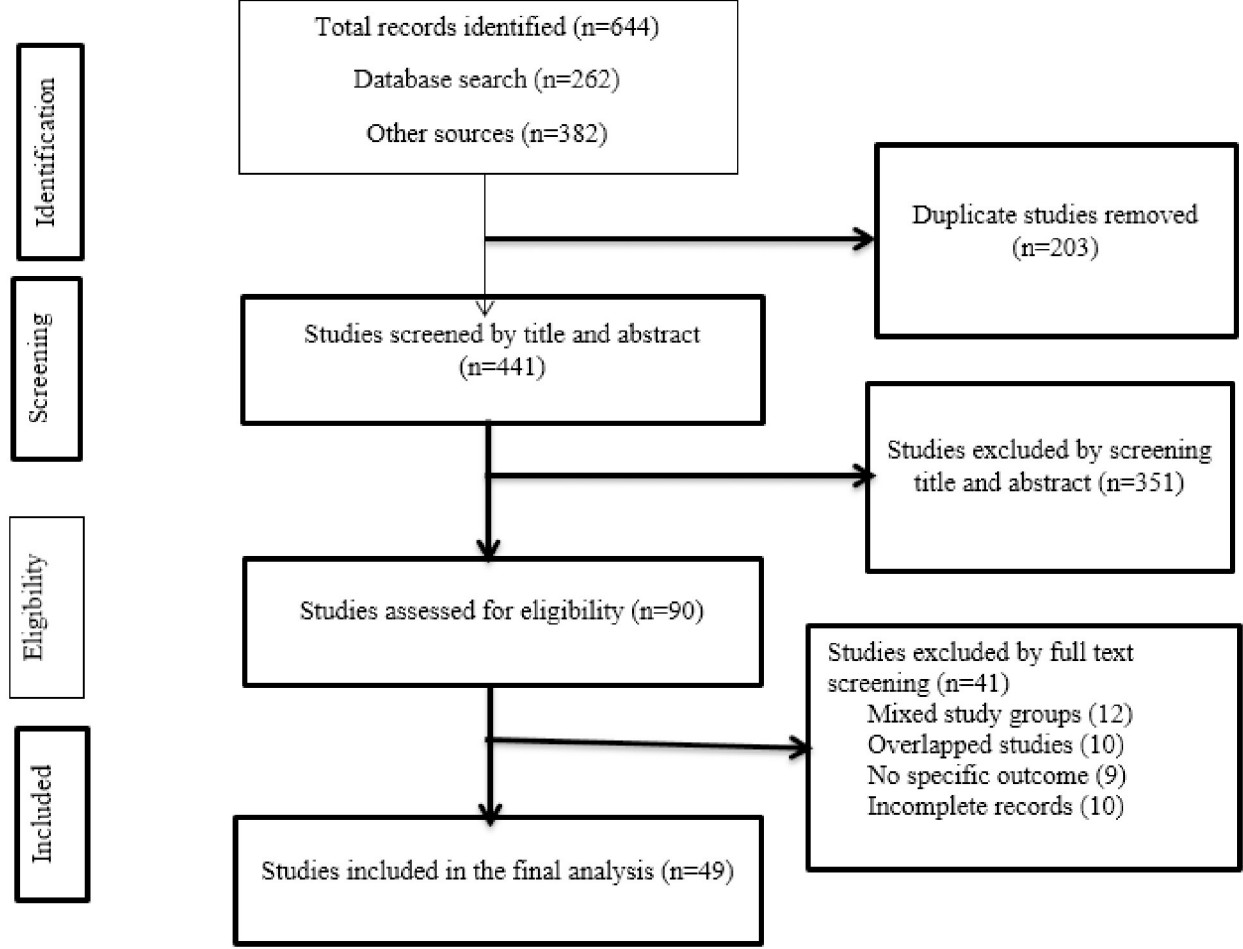

**Fig 1. Flowchart diagram describing a selection of studies for the systematic review and meta-analysis on the predictors of mortality in patients with drug-resistant tuberculosis.**

## Quality assessment

We evaluated the quality of eligible articles using the Joanna Briggs Institute Critical Appraisal (JBI) tools designed for case-control and cohort studies [62]. The cohort checklist consists of 11 indicators and the case-control checklist consists of 10 indicators. These indicators were turned into 100% and the quality score was graded as high if >80%, a medium between 60–80%, and low <60%. Two authors (AA[1] and DFG) conducted the quality assessment, and the third author TW managed the inconsistencies (S4 Table).

## Outcomes

Mortality from any cause in patients with DR-TB during their anti-TB treatment course was the primary outcome. Predictors of mortality were the second outcome. The pooled HR, RR, and OR along with their 95% CIs were estimated to assess these predictors of mortality in patients with DR-TB.

## Data analysis

Data extracted in Microsoft excel 2016 were imported into STATA Version 15 for analysis. We estimated the proportion, incidence, and predictors of mortality in patients with DR-TB.

**Table 1. Characteristics of individual studies on the predictors of mortality in patients with drug-resistant tuberculosis, included in the current systematic review and meta-analysis.**

| Author, Year | Study country | Study design | Study period | Study age-group | Study setting | Sample size | Number of deaths | Death | | Quality score |
|---|---|---|---|---|---|---|---|---|---|---|
| | | | | | | | | Proportion | Incidence density | |
| Bajehson et al., 2019 [10] | Nigeria | CC | 2015–16 | All | Kano, Katsina and Bauchi states of Nigeria | 147 | 38 | 25.85% | - | High |
| Balabanova et al., 2016 [11] | Latvia, Lithuania, Estonia and Bucharest city | RC | 2007–12 | All | National TB and Infectious Diseases University Hospital in Vilnius, Clinic of TB and Lung Diseases at Riga East University hospital, Lung Hospital at Tartu University, Estonia and Marius Nasta Institute of Pneumology, Bucharest, Romania. | 737 | 227 | 30.80% | 3.00 per 10,000 days | High |
| Bei et al., 2018 [12] | China | RC | 2013–17 | All | Changsha Central Hospital, Wuhan Treatment Center, the Third People's Hospital of Hengyang, and the Second People's Hospital of Chenzhou | 67 | 20 | 29.85% | 3.51 per 10,000 days | High |
| Bhering et al., 2019 [19] | Brazil | RC | 2000–16 | All | Tuberculosis Surveillance System in Rio de Janeiro State | 2269 | 1,005 | 44.29% | - | High |
| Brust et al., 2018 [20] | South Africa | RC | 2011–13 | All | KwaZulu-Natal province | 191 | 24 | 12.57% | - | High |
| Chingonzoh et al., 2018 [13] | South Africa | RC | 2011–13 | ≥18 Yrs | Registered on the routine DR-TB reporting database in the Eastern Cape Province | 3,729 | 1,445 | 38.75% | - | High |
| Delgado et al., 2015 [21] | Peru | RC | 2000–12 | ≥18 Yrs | Clinical records of the National Strategy for Prevention and Control of Tuberculosis in Lima | 236 | 44 | 18.64% | - | High |
| Dheda et al., 2010 [22] | South Africa | RC | 2002–08 | >16 Yrs | Four (Western Cape, Eastern Cape Gauteng Northern Cape) dedicated provincial facilities for the treatment of XDR tuberculosis | 174 | 62 | 35.63% | - | High |
| Fantaw et al., 2018 [23] | Ethiopia | RC | 2013–17 | All | Adama and Bishoftu General Hospitals | 164 | 30 | 18.29% | 4.75 per 10,000 days | High |
| Farley et al., 2011 [24] | South Africa | RC | 2000–04 | ≥18 Yrs | Ten participating MDR-TB treatment centers from eight South African provinces | 757 | 177 | 23.38% | - | Medium |
| Gandhi et a., 2012 (MDR-TB) [25] | South Africa | CC | 2005–06 | All | Tugela Ferry | 123 | 78 | 63.41% | - | High |
| Gandhi et a., 2012 (XDR-TB) [25] | | | | | | 139 | 111 | 79.86% | - | High |
| Gayoso et al., 2018 [26] | Brazil | RC | 2005–12 | All | HélioFraga Reference Center (ENSP-FIOCRUZ) | 3802 | 479 | 12.60% | - | High |
| Gebre et al., 2020 [27] | Ethiopia | RC | 2012–17 | Adults | Dil Chora Referral Hospital, Amir Nur Health Center, and Hailemariam Referral Hospital. | 362 | 55 | 15.19% | 4.14 per 10,000 days | High |
| Getachew et al., 2013 [28] | Ethiopia | RC | 2009–12 | All | St. Peter's Specialized Tuberculosis Hospital | 188 | 29 | 15.43% | 3.64 per 10,000 days | High |
| Girum et al, 2017 [8] | Ethiopia | RC | 2013–17 | All | Yirgalem and Queen Eleni Memorial Hospital | 154 | 13 | 8.44% | 1.91 per 10,000 days | High |
| Janmeja et al., 2018 [29] | India | RC | 2012–14 | All | Department of Pulmonary Medicine, Government Medical College, and Hospital, Chandigarh. | 278 | 61 | 21.94% | - | High |

(*Continued*)

**Table 1.** (Continued)

| Author, Year | Study country | Study design | Study period | Study age-group | Study setting | Sample size | Number of deaths | Death | | Quality score |
|---|---|---|---|---|---|---|---|---|---|---|
| | | | | | | | | Proportion | Incidence density | |
| Jeon et al., 2011 [30] | South Korea | RC | 2004 | ≥16Yrs | National Mokpo Tuberculosis Hospital, Mokpo, National Masan Tuberculosis Hospital, Masan, and Seobuk Hospital, Seoul, Korea | 202 | 127 | 62.87% | - | High |
| Kang et al., 2013 [31] | South Korea | RC | 2000–02 | ≥20 Yrs | All national TB hospitals (n = 360), all Korean National Tuberculosis Association (KNTA) chest clinics (n = 836) and eight randomly selected university hospitals near Seoul (n = 211). | 1,407 | 470 | 33.40% | - | High |
| Kanwal et al., 2017 [9] | Pakistan | RC | 2010–15 | All | 11 programmatic management of DR-TB centers in Punjab | 1,136 | 472 | 41.55% | - | Medium |
| Kassa et al., 2020 [32] | Ethiopia | RC | 2010–2017 | All | University of Gondar, Borumeda, and Debre-Markos Referral Hospital | 451 | 46 | 10.20% | 2.03 per 10,000 days | High |
| Kashongwe et al, 2017 [33] | Democratic Republic of Congo | RC | 2015–17 | All | Kinshasa TB Referral Hospital | 119 | 18 | 15.13% | - | High |
| Kim et al., 2010 [34] | South Korea | RC | 2000–02 | ≥13 Yrs | Registry of the Korea National Statistical Office | 1407 | 144 | 10.23% | - | High |
| Kizito et al., 2021 [35] | Uganda | CC | 2016 | All | National MDR-TB cohort. | 198 | - | - | - | High |
| Kurbatova et al., 2012 [36] | Estonia, Latvia, Philippine, Russia, Peru | RC | 2000–04 | Adults | DOTS-Plus programs | 1768 | 200 | 11.31% | - | High |
| Makhmudova et al., 2019 [37] | Tajikistan | RC | 2012–13 | ≥18 Yrs | 32 of 37 TB facilities in the selected districts. | 601 | 89 | 14.81% | - | High |
| Manda et al, 2014 [38] | South Africa | RC | 2000–14 | All | Standardized Programmatic Management of MDR-TB | 1619 | 367 | 22.67% | - | High |
| Milanov et al., 2015 [39] | Bulgaria, | RC | 2009–10 | ≥18 Yrs | Hospital for Lung Diseases in Gabrovo and the TB registers of the NRL-TB at the NCIPD in Sofia | 50 | 19 | 38.00% | - | High |
| Mitnick et al., 2013 [40] | Peru | RC | 1999–02 | All | All patients who were enrolled between 1 February 1999 and 31 July 2002 in Lima, Peru, in ambulatory treatment for MDR-TB, | 669 | 139 | 20.78% | - | High |
| Molalign et al., 2015 [41] | Ethiopia | RC | 2011–14 | All | ALERT and Gondar University Teaching and Referral Hospital | 342 | 37 | 10.82% | 2.33 per 10,000 days | High |
| Mollel et al., 2017 [42] | Tanzania | RC | 2012–14 | All | Kibong'oto Infectious Diseases Hospital (KIDH) | 193 | 13 | 6.74% | - | High |
| O'Donnell et al., 2013 [43] | South Africa | RC | 2006–2009 | ≥18 years | Public TB referral hospital in KwaZulu-Natal Province | 114 | 48 | 42% | - | High |
| Olaleye et al., 2016 [44] | South Africa | RC | 2001–10 | ≥15 Yrs | A specialized TB hospital in Witbank | 442 | 151 | 34.16% | 8.18 per 10,000 day | High |
| Park et al., 2010 [45] | South Korea | RC | 2004 | All | 21 private hospitals | 170 | 12 | 7.06% | - | High |
| Pradipta et al., 2019 [46] | Netherlands | RC | 2005–15 | Adults | Nationwide exhaustive registry of tuberculosis patients | 103 | 3 | 2.91% | - | High |
| Prajapati et al., 2017 [47] | Gujarat, India | PC | 2012–16 | All except pregnant | B. J. Medical College, Civil Hospital | 112 | 58 | 51.79% | - | High |
| Rusisiro et al., 2019 [48] | Rwanda | RC | 2014–17 | All | Rwanda National Tuberculosis Program: DR-TB excel database. | 279 | 31 | 11.11% | - | High |

(*Continued*)

**Table 1.** (Continued)

| Author, Year | Study country | Study design | Study period | Study age-group | Study setting | Sample size | Number of deaths | Death Proportion | Death Incidence density | Quality score |
|---|---|---|---|---|---|---|---|---|---|---|
| Samali et al, 2017 [49] | Tanzania | RC | 2009–2016 | All | Kibong'oto hospital | 583 | 89 | 15.27% | 4.80 per 10,000 days | High |
| Schnippel et al., 2015 [50] | South Africa | RC | 2009–11 | All | Electronic Drug-Resistant Tuberculosis Register by National TB Programme | 10,763 | 2,987 | 27.75% | - | High |
| Seifert et al., 2017 [51] | India, Moldova and South Africa | PC | 2012–13 | All | Selected hospitals and clinics with a high prevalence of drug-resistant TB in India, Moldova, and South Africa. | 834 | 62 | 7.43% | 3.91 per 10,000 days | High |
| Seung et al., 2009 [52] | Lesotho | RC | 2007–08 | All | Lesotho national MDR-TB program | 76 | 22 | 28.95% | - | High |
| Shariff et al., 2016 [53] | Malaysia | RC | 2009–13 | All | Patients receiving treatment at the Institute of Respiratory Medicine in Kuala Lumpur | 426 | 65 | 15.26% | 1.40 per 10,000 days | High |
| Shenoi et al., 2012 [54] | South Africa | CC | 2005–08 | All | Tugela Ferry | 142 | 73 | 51.41% | - | High |
| Shimbre et al., 2020 [55] | Ethiopia | RC | 2009–16 | All | Dile Chora, Yirgalem, Queen Eleni Mohamed Memorial and Shene Gibe Hospitals | 462 | 38 | 8.23% | 1.86 per 10,000 days | High |
| Sun et al., 2015 [56] | China | RC | 2001–02 | All | Henan Province | 86 | 37 | 43.02% | 1.07 per 10,000 days | High |
| Suryawanshi et al., 2017 [57] | India | RC | 2011–12 | All | PMDT records in Maharashtra state | 3410 | 857 | 25.13% | - | High |
| Wai et al., 2017 [58] | Myanmar | RC | 2015–17 | All | Community-Based in 33 townships of upper Myanmar | 261 | 26 | 9.96% | 5.50 per 10,000 days | High |
| Wang et al., 2019 [59] | China | RC | 2006–14 | All | TB management information system | 552 | | | - | High |
| Wang et al., 2020 [60] | China | RC | 2006–2011 | Adult | Wuhan Pulmonary Hospital | 356 | 103 | 28.93% | 10.44 per 10,000 days | High |
| Woya et al., 2019 [61] | Ethiopia | RC | Up to Feb 2018 | All | Different MDR-TB Hospitals of Amhara Region | 207 | 61 | 29.47% | 3.08 per 10,000 days | High |

CC; Case-control, MDR-TB; Multi-Drug Resistant Tuberculosis, PC; Prospective Cohort, RC; Retrospective Cohort, TB; Tuberculosis, XDR-TB; Extensively Drug-resistant tuberculosis.

The proportion was estimated by dividing the number of deaths by the total sample size, while the incidence rate was described per 10,000 person-days of follow-up. To assess the predictors, the pooled HR, RR, and OR with 95% CI were estimated by assuming the true effect size varies between studies. For studies that did not present the measures of association, we analyzed the estimates along with 95% CI. We presented the meta-analysis results using a forest plot. Also, we assessed the heterogeneity among the studies using the forest plot and $I^2$ heterogeneity test [63]. We used a fixed-effects model for $I^2 < 50\%$ and a random-effects model for $I^2 > 50\%$ to perform the analysis [64]. Besides, publication bias was explored using visual inspection of the funnel plot, and Egger's regression test was carried out to check the statistical symmetry of the funnel plot.

## Role of the funding source

No fund was obtained to execute this systematic review and meta-analysis.

**Table 2. The summary of the pooled estimates of the HR, RR, and OR per predicting factors of mortality in patients with drug-resistant tuberculosis.**

| Variable | HR | | | | RR | | | | OR | | | |
|---|---|---|---|---|---|---|---|---|---|---|---|---|
| | Number of studies | Estimate, 95%CI | Heterogeneity | | Number of studies | Estimate, 95%CI | Heterogeneity | | Number of studies | Estimate, 95% CI | Heterogeneity | |
| | | | I² | P-value | | | I² | P-value | | | I² | P-value |
| Adverse effect | 5 | 0.70 (0.44,0.96) | 69.3% | 0.011 | NA | NA | NA | NA | NA | NA | NA | NA |
| Alcohol | 8 | 1.19 (0.65,1.73) | 60.5% | 0.013 | 3 | 1.87 (0.98,2.76) | 73.6% | 0.023 | 4 | 1.59 (0.28,2.91) | 43.6% | 0.150 |
| Anemia | 7 | 1.79 (0.98,2.59) | 84.3% | <0.001 | NA | NA | NA | NA | 2 | 3.56 (0.07,7.06) | 0.0% | 0.841 |
| BMI<18.5 | 12 | 1.62 (1.28,1.97) | 87.2% | <0.001 | 3 | 3.13 (2.17,4.09) | 0.0% | 0.608 | 2 | 2.79 (0.31,13.50) | 0.0% | 0.334 |
| Cavitation | 5 | 1.16 (0.88,1.44) | 61.6% | 0.034 | 4 | 1.04 (0.72,1.36) | 51.3% | 0.104 | 6 | 0.78 (0.57,0.98) | 37.7% | 0.155 |
| Any comorbidity | 19 | 1.92 (1.50,2.35) | 61.4% | <0.001 | 6 | 1.61 (1.29,1.93) | 0.0% | 0.543 | 6 | 1.58 (1.09,2.06) | 0.0% | 0.730 |
| Diabetes | 9 | 1.74 (1.24,2.24) | 37.3% | 0.120 | 2 | 1.60(1.13, 2.07) | 0.0% | 0.385 | 3 | 0.72(0.40, 1.04) | 0.0% | 0.722 |
| EPTB involvement | 5 | 1.52 (0.96,2.08) | 66.0% | 0.019 | 7 | 0.96(0.47– 1.46) | 90.7% | <0.001 | 5 | 0.71(-0.05, 1.48) | 44.0% | 0.129 |
| HIV co-infection | 16 | 2.15(1.69, 2.61) | 48.2% | 0.016 | 12 | 1.49(1.27, 1.72) | 19.5% | 0.253 | 13 | 1.62(1.41, 1.84) | 0.0% | 0.549 |
| Male sex | 17 | 1.25 (1.08,1.41) | 30.5% | 0.113 | 12 | 0.93(0.88, 0.98) | 35.3% | 0.108 | 14 | 0.76(0.62, 0.90) | 13.5% | 0.306 |
| Older age | 17 | 2.13(1.64, 2.62) | 59.0% | 0.001 | 6 | 1.40(1.26, 1.53) | 48.4% | 0.084 | 10 | 1.51(0.95, 2.07) | 50.3% | 0.034 |
| Previous TB history | 18 | 1.30(1.06, 1.54) | 64.6% | <0.001 | 6 | 1.12(0.63, 1.61) | 98.3% | <0.001 | 8 | 1.41(0.43, 2.38) | 95.5% | <0.001 |
| Previous SLD treatment | 5 | 2.52 (2.15,2.88) | 0.0% | 0.706 | 2 | 1.06(-0.05, 2.16) | 93.7% | <0.001 | 3 | 1.38(0.29, 2.46) | 94.6% | <0.001 |
| Smear positive at baseline | 7 | 1.45 (1.14,1.76) | 49.2% | 0.066 | 3 | 1.58(1.46, 1.69) | 48.7% | 0.142 | 2 | 5.33(1.31, 9.36) | 0.0% | 0.639 |
| Smoking | 10 | 1.14 (0.70,1.59) | 51.6% | 0.029 | 6 | 1.29(0.61, 1.97) | 81.1% | <0.001 | 6 | 0.81((0.17, 1.44) | 42.6% | 0.121 |
| Substance addiction | 5 | 1.44(0.57, 2.32) | 80.8% | <0.001 | NA | NA | NA | NA | NA | NA | NA | NA |
| Treatment delay | 2 | 1.57(-0.39, 3.53) | 37.7% | 0.205 | 4 | 1.12 (0.65,1.59) | 62.9% | 0.044 | NA | NA | NA | NA |
| XDR-TB | 7 | 2.01(1.50, 2.52) | 60.8% | 0.018 | 3 | 2.44(2.16, 2.73) | 46.1% | 0.156 | 5 | 2.21(1.05, 3.37) | 51.2% | 0.084 |
| Clinical complication | 8 | 2.98(2.32, 3.64) | 69.9% | 0.002 | NA | NA | NA | NA | NA | NA | NA | NA |
| For a one year increase in age | 10 | 1.01(1.00, 1.03) | 73.0% | <0.001 | NA | NA | NA | NA | NA | NA | NA | NA |

BMI; Body Mass Index, EPTB; Extra Pulmonary Tuberculosis, HR; Hazards Ratio, NA; Not applicable, OR; Odd's Ratio, RR; Risk Ratio, SLD; Second-line Drugs, TB; Tuberculosis, XDR-TB; Extensively Drug-resistant tuberculosis.

## Results

### Study characteristics

From the whole search, we assessed 640 articles for eligibility. After 203 studies were removed by duplication, 437 articles were screened by title and abstract. Then, 351 articles were excluded and full-text screening was conducted on 86. Accordingly, 41 studies were excluded

from the study due to mixed study groups (12), overlapped studies (10), did not have specific outcomes (9), and incomplete records (9). Therefore, in this systematic review and meta-analysis, 49 studies [8–13, 19–61] were included in the final analysis (Fig 1). These 49 individual studies were conducted in 25 different countries located in four continents (Africa, Asia, Europe, and South America). More than half (25, 51%) of the included studies were from Africa. The remaining 15 studies were from Asia; four from South America, three from Europe, and two from multi-center studies. South Africa (11 studies) and Ethiopia (8 studies) contributed to the large proportion of individual studies included in the current study followed by India (4 studies) and South Korea (4 studies). The majority of the primary studies were based on data collected from patients enrolled in hospitals for treatment. Data were collected either directly from patient registries or the health information system (national or regional database) or the prospective cohort research project database. Most (45, 91.8%) of the studies used a retrospective cohort study design: however, some studies also used either a case-control or prospective cohort study design. The study period ranged from 1999 to 2017. Besides, most of the studies were conducted on all age groups (Table 1). Based on the results found through the JBI quality assessment tool, the indicators were turned in to 100% and graded as high if >80%, medium between 60–80%, and low <60%. Accordingly, the majority of the studies (47 out of 49) were graded to have high quality, and only two studies were categorized under medium quality (S4 Table).

## Proportion and incidence of death

The smallest sample size was 67 in a study done by Bei et al., 2018 [12], while the largest sample size was 10,763 in a study done by Schnippel et al., 2015 [50]. We estimated the pooled proportion of mortality and the incidence of mortality in patients with DR-TB based on 48 study results and 17 studies respectively. The proportion of death ranged from 6.74% [42] to 79.86% [25], while the incidence of mortality ranged from 1.07 per 10,000 person-days [56] to 10.44 per 10,000 person-days [60]. Based on the random-effects model, the proportion of death and the incidence of mortality in patients with DR-TB during their treatment follow-up period were 25.62% (95%CI; 20.91, 30.33, $I^2$; 99.31%) (Fig 2), and 3.75 per 10,000 person-days (95% CI; 2.65, 4.86, $I^2$; 97.61%), respectively (Fig 3). We evaluated the publication bias using the visual inspection of the funnel plot and Egger's test. Accordingly, the funnel plot revealed that there was no publication bias, and the symmetry of the funnel plot was confirmed by a non-significant Egger's test result (death incidence, *p = 0.465*) (Fig 4), (death incidence density rate, *p = 0.051*) (Fig 5). The sensitivity analysis was done for the pooled incidence of death, and it was found that no single study affected the pooled death incidence. In this study, we separately analyzed the mortality incidence for MDR-TB patients and XDR-TB patients. Accordingly, the proportion of death in patients with MDR-TB and XDR-TB during their treatment follow-up period was 20.21% (95%CI; 16.45, 23.97, $I^2$; 98.76%) (Fig 6), and 43.53% (95%CI; 35.08, 51.97, $I^2$; 96.29%) respectively (Fig 7). Besides, the funnel plot revealed that there was no publication bias both for MDR-TB (Fig 8) and for XDR-TB (Fig 9).

## Predictors of mortality

We assessed the pooled estimate for different predictors. The predictors included demographic (sex and age), behavioral (alcohol use, smoking and substance addiction) and clinical (adverse effect, anemia, undernutrition, comorbidities, diabetes, EPTB involvement, HIV sero-status, cavitation, previous TB history, previous SLD treatment, clinical complication, treatment delay, smear positive TB and drug resistance pattern) characteristics. Based on the pooled analysis of the hazards ratio and risk ratio, under nutrition (HR = 1.62,95%CI;1.28,1.97, $I^2$;87.2%,

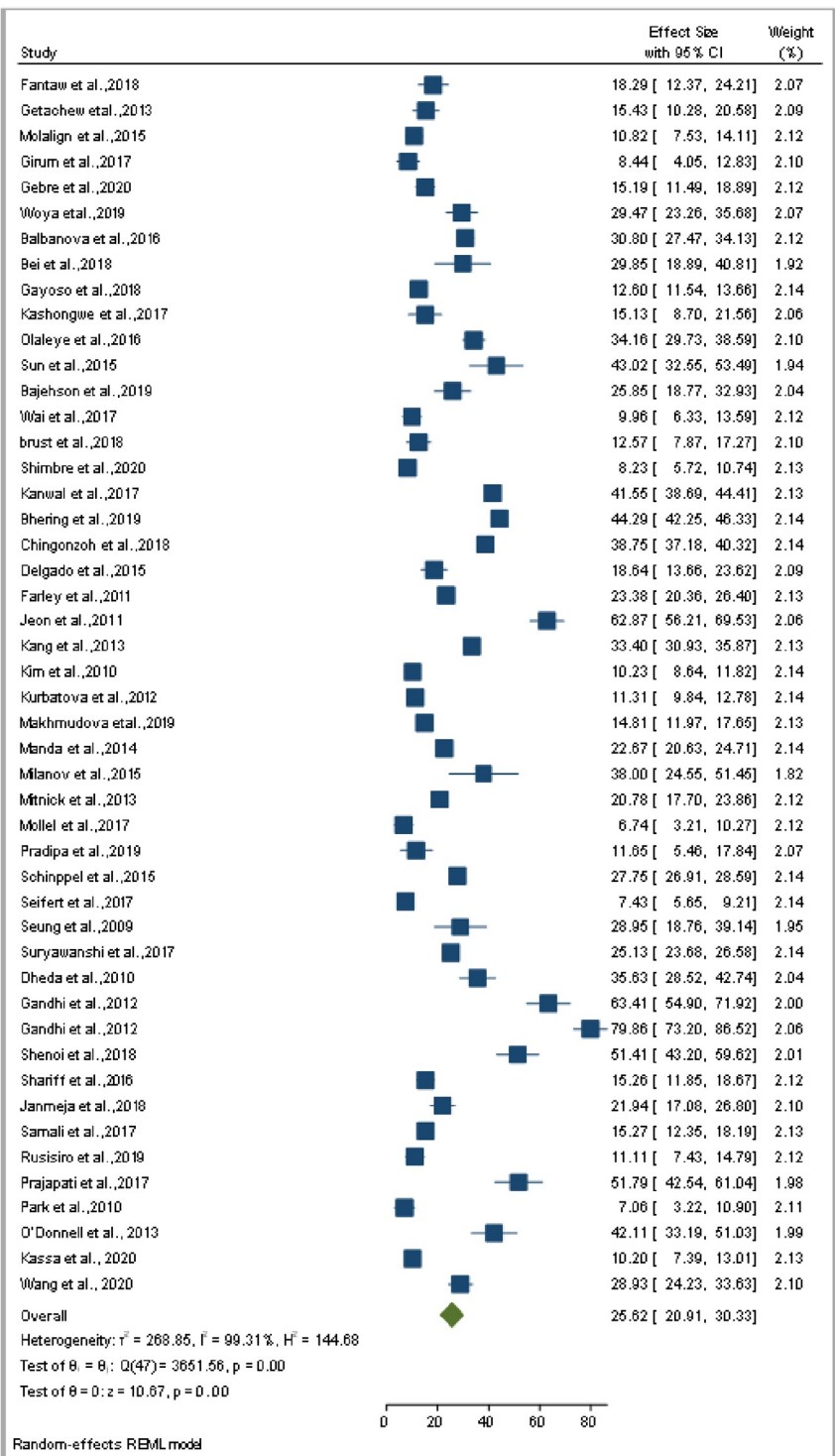

**Fig 2. Forest plot for pooled incidence of mortality in patients with drug-resistant tuberculosis.**

RR = 3.13, 95% CI;2.17,4.09 $I^2$; 0.0%), presence of any type of co-morbidity (HR = 1.92,95% CI;1.50–2.33,$I^2$;61.4%, RR = 1.61, 95% CI; 1.29,1.93, $I^2$; 0.0%), having diabetes (HR = 1.74, 95% CI; 1.24,2.24, $I^2$;37.3%, RR = 1.60, 95%CI; 1.13, 2.07, $I^2$; 0.0%), HIV co-infection (HR = 2.15, 95%CI;1.69,2.61, $I^2$; 48.2%, RR = 1.49, 95%CI;1.27, 1.72, $I^2$; 19.5%), male sex (HR = 1.25,95%

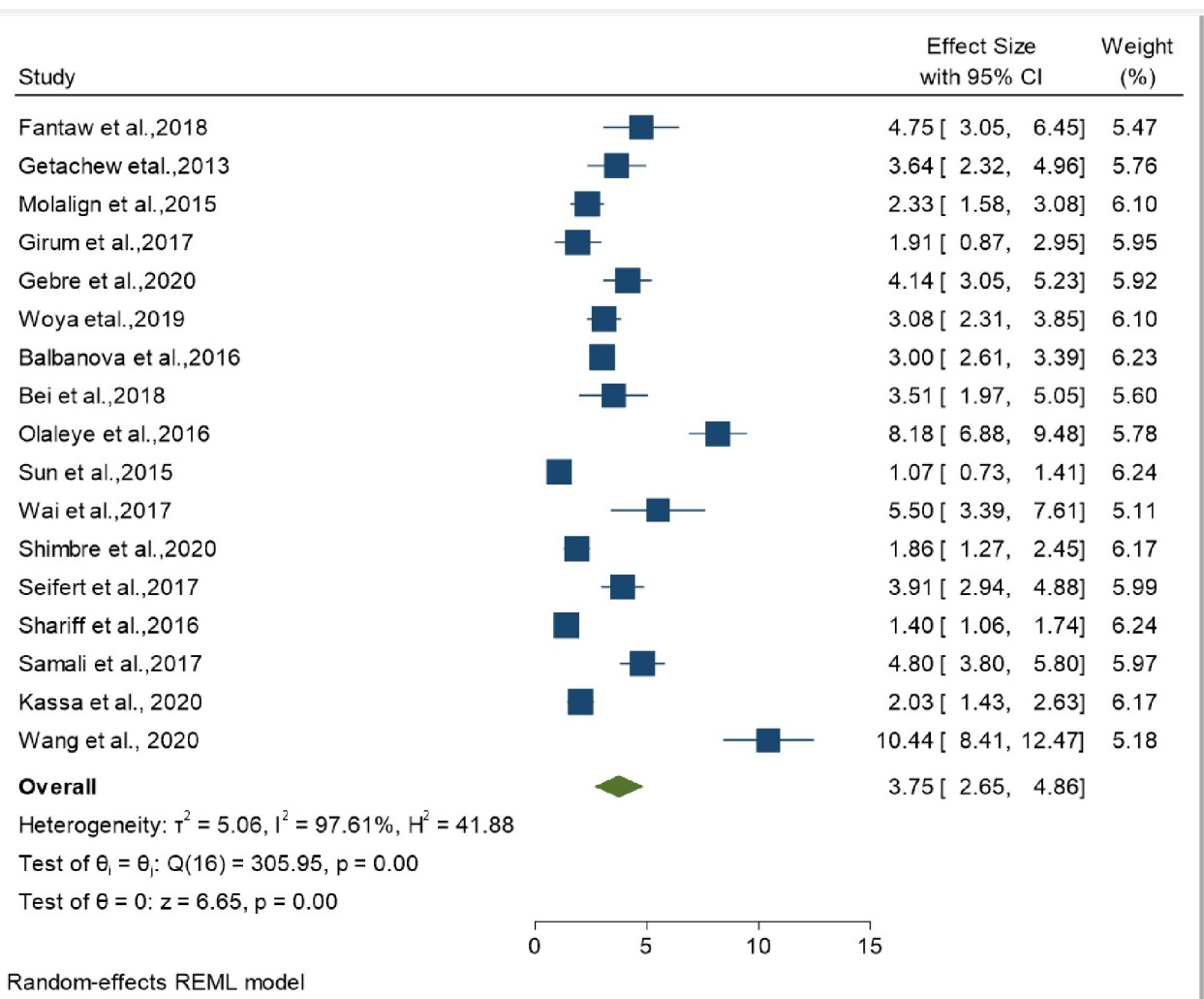

**Fig 3. Forest plot for pooled incidence density rate of mortality in patients with drug-resistant tuberculosis.**

CI;1.08,1.41,$I^2$;30.5%), older age (HR = 2.13, 95%CI;1.64,2.62,$I^2$;59.0%,RR = 1.40,95%CI; 1.26, 1.53, $I^2$; 48.4%) including a 1 year increase in (HR = 1.01, 95%CI;1.00,1.03,$I^2$;73.0%), previous TB history (HR = 1.30,95%CI;1.06,1.54, $I^2$;64.6%), previous second line anti-TB treatment (HR = 2.52, 95%CI; 2.15,2.88, $I^2$; 0.0%), being smear positive at the baseline (HR = 1.45, 95% CI;1.14,1.76, $I^2$;49.2%, RR = 1.58,95%CI;1.46,1.69, $I^2$;48.7%), having XDR-TB (HR = 2.01, 95% CI;1.50, 2.52, $I^2$; 60.8%,RR = 2.44, 95% CI;2.16, 2.73, 46.1%), and any type of clinical complication (HR = 2.98,95%CI; 2.32, 3.64, $I^2$; 69.9%) were the predictors of mortality in patients with DR-TB (Fig 10). Also, our pooled analysis of the odds ratio showed that any cause of mortality in patients with DR-TB is associated with the presence of any type of comorbidity (OR = 1.58, 95%CI;1.09,2.06, $I^2$; 0.0%), HIV co-infection (OR = 1.62, 95%CI;1.41, 1.84, $I^2$; 0.0%), being smear positive at the baseline (OR = 5.33, 95%CI;1.31, 9.36 $I^2$; 0.0%), and having XDR-TB (OR = 2.21, 95%CI = 1.05, 3.37, $I^2$; 51.2%) (Table 2).

However, statistically significant differences or associations was not observed for alcohol consumption (HR = 1.19, 95% CI; 0.65,1.73, $I^2$; 60.5%RR = 1.87,95%CI;0.98,2.76, $I^2$; 73.6%), presence of any grade of anemia (HR = 1.79,95%CI = 0.98,2.59, $I^2$;84.3%), presence of

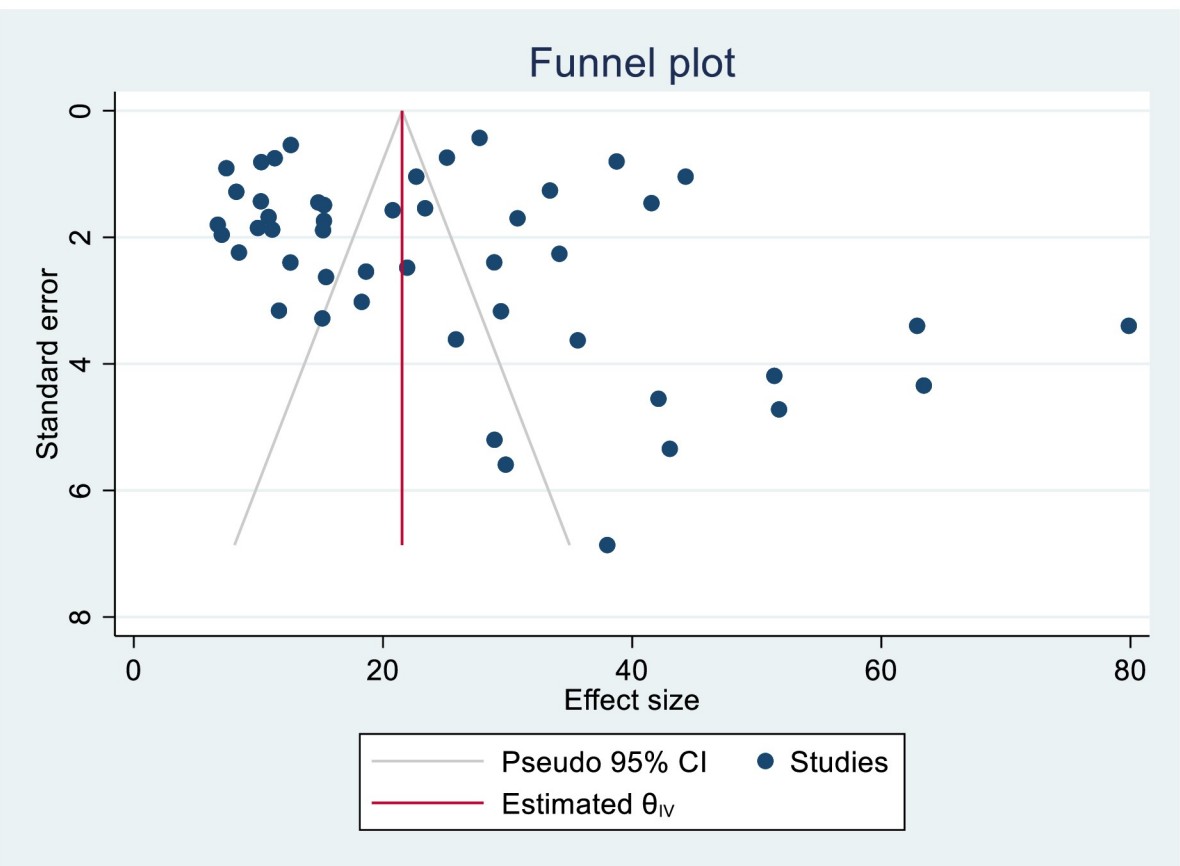

**Fig 4. Funnel plot showing publication bias among studies used to compute the incidence of mortality in patients with drug-resistant tuberculosis.**

cavitation (HR = 1.16%CI; 0.88–1.44, $I^2$;61.6%, RR = 1.04, 95%CI;0.72,1.36, $I^2$; 51.3%), extra-pulmonary involvement (HR = 1.52, 95%CI;0.96,2.08, $I^2$;66.0%,RR = 0.96, 95%CI;0.47–1.46, $I^2$; 90.7%), smoking (HR = 1.14, 95%CI; 0.70,1.59, $I^2$;51.6%, RR = 1.29, 95% CI;0.61, 1.97, $I^2$; 81.1%), addiction to substances (HR = 1.44, 95%CI;0.57, 2.32, $I^2$; 80.8%), and treatment delay (HR = 1.57, 95%CI;-0.39, 3.53, $I^2$; 37.7%, RR = 1.12, 95%CI; 0.65,1.59, $I^2$; 62.9%) (Table 2).

## Predictors of mortality per drug-resistant tuberculosis categories

In the current study, we performed a sub-group analysis to assess the predictors of mortality based on the drug-resistance category of TB. As per the presentation in the studies included in this study, the resistance category is classified as; 1) RR/MDR 2) mixed MDR and XDR 3) XDR 4) DR-TB. However, the fourth category (DR-TB) is not specified due to the mix-up of different resistance included in the individual studies. Thus, we presented based on the data available in the specific studies; A) DR-TB; Poly DR, RR, MDR, XDR B) DR-TB; Poly DR, MDR, XDR C) DR-TB; RIF/INH-mono resistant, MDR, XDR D) DR-TB; mono resistance, poly resistance, MDR E) DR-TB; mono resistance, poly resistance, MDR, XDR.

Specific to the type of DR-TB, the predictors of mortality among RR/MDR patients includes; older age (HR = 1.72, 95%CI;1.15, 2.29, $I^2$; 31.5%), one year increase in age (HR = 1.01, 95%CI;1.00, 1.02, $I^2$; 73.7%), presence of any type of co-morbidity (HR = 2.39,

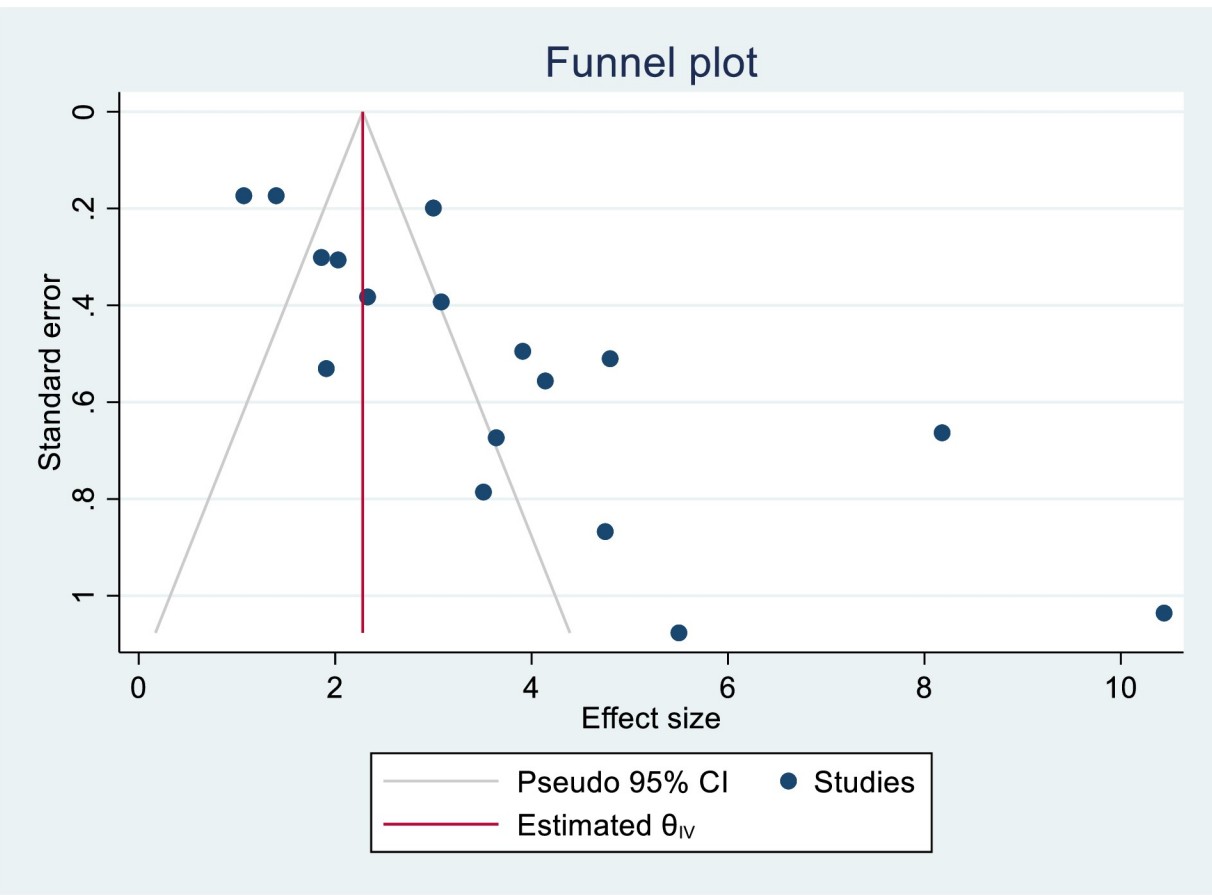

**Fig 5. Funnel plot showing publication bias among studies used to compute the incidence density rate of mortality in patients with drug-resistant tuberculosis.**

95%CI;1.57, 3.21, $I^2$; 72.2%), having diabetes (HR = 2.05, 95%CI;1.40,2.70, $I^2$;0.00%), HIV co-infection (HR = 2.35, 95%CI;1.68,2.82, $I^2$;24.1%), previous TB history (HR = 1.46, 95%CI;1.19, 1.72, $I^2$;0.00%), being smear-positive at the baseline (HR = 3.05, 95%CI; 2.17, 4.29), and any type of clinical complication (HR = 2.80, 95%CI; 1.86, 3.74, $I^2$; 62.1%). While, the predictors of mortality among mixed MDR and XDR-TB patients includes; being male (HR = 1.52, 95% CI;1.30, 1.73, $I^2$; 0.00%), older age (HR = 2.39, 95%CI;1.75, 3.03, $I^2$; 61.3%), one year increase in age (HR = 1.02, 95%CI;1.01, 1.04, $I^2$; 0.00%), undernutrition (HR = 2.33, 95%CI;1.19, 3.47, $I^2$; 94.1%), presence of any type of co-morbidity (HR = 1.87, 95%CI;1.36, 2.37, $I^2$; 33.8%), previous second-line anti-TB treatment (HR = 2.50, 95%CI;2.13,2.87, $I^2$;0.00%), and being smear-positive at the baseline (HR = 1.35, 95%CI; 1.16, 1.53, $I^2$; 0.00%). Among XDR-TB patients the predictors of mortality were; older age (HR = 2.82, 95%CI; 1.08, 7.35, single study), undernutrition (HR = 4.30, 95%CI; 1.26, 14.72), presence of any type of co-morbidity (HR = 5.16, 95% CI; 2.05, 13.00), and previous second-line anti-TB treatment (HR = 3.73, 95%CI; 1.69, 8.22). Besides, among DR-TB patients with a mix of different resistance categories, the predictors of mortality were as follows. Among DR-TB patients with a mix of mono drug-resistant, poly drug-resistant, and MDR-TB, older age is a predictor of mortality (HR = 5.29, 95%CI; 1.02, 27.29). While among DR-TB patients with a mix of RIF/INH-mono drug-resistant, MDR, and XDR-TB, a one-year increase in age (HR = 1.02, 95%CI; 1.01, 2.20), and undernutrition (HR = 1.96, 95%CI; 1.15, 3.35) were the predictors of mortality. The presence of any type of

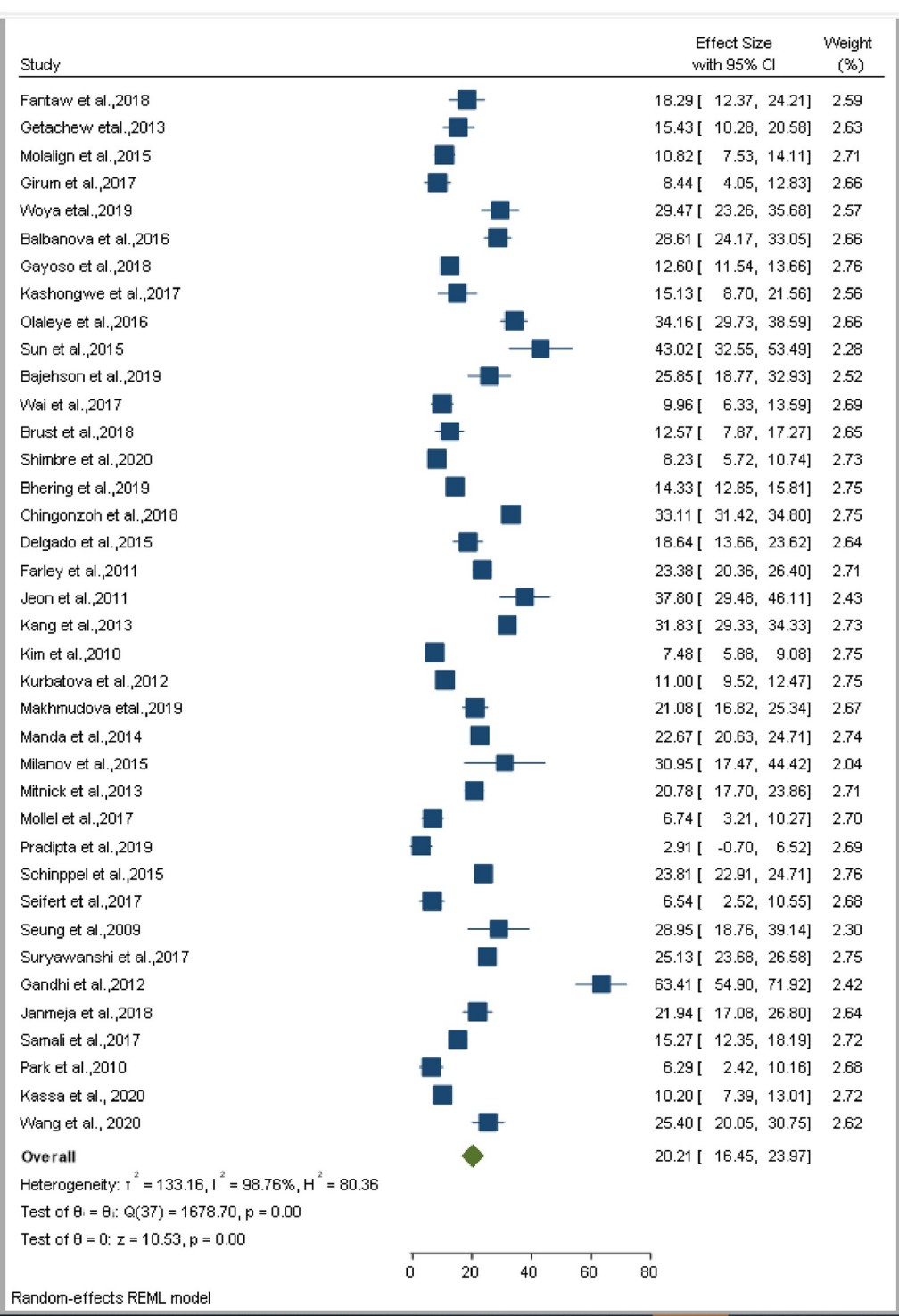

**Fig 6. Forest plot for pooled incidence of mortality in patients with multi drug-resistant tuberculosis.**

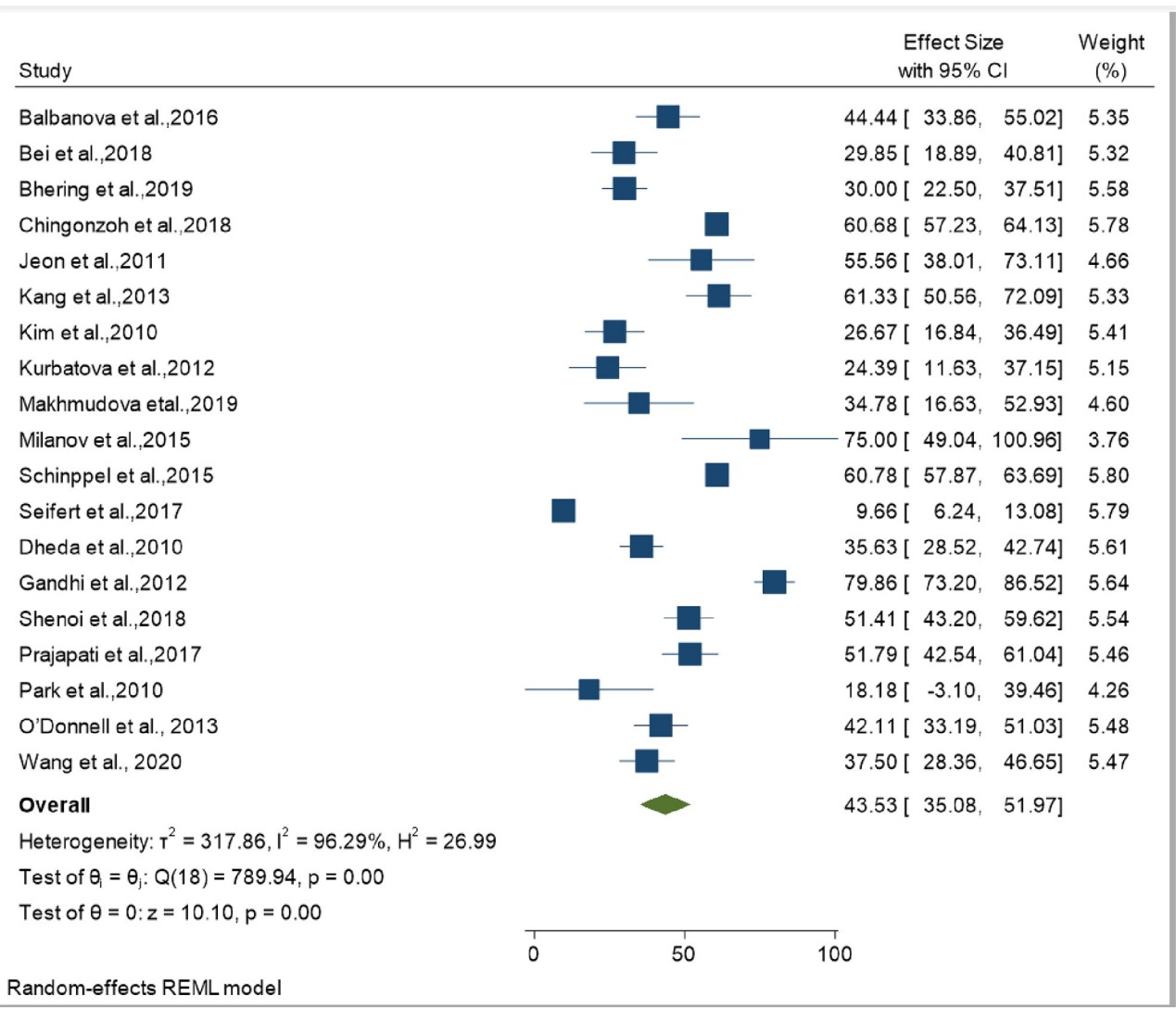

**Fig 7. Forest plot for pooled incidence of mortality in patients with extensively drug-resistant tuberculosis.**

co-morbidity is a predictor of mortality among DR-TB patients with a mix of poly drug-resistant, RR, MDR, and XDR-TB (HR = 1.34, 95%CI; 1.03, 1.74), and among DR-TB patients with a mix of mono drug-resistant, poly drug-resistant, and MDR-TB (HR = 8.44, 95%CI; 3.04, 23.44) (Fig 10).

## Discussion

In this systematic review and meta-analysis, we analyzed the pooled data to assess the predictors of mortality in patients with DR-TB based on studies conducted in different countries and settings at the global level. The case definition for drug-resistant TB in this study was according to the WHO definition such that any TB case caused by *Mycobacterium tuberculosis* resistant to at least one anti-TB drug. Based on the pooled estimates, the predictors of mortality include: male sex, older age, undernutrition, HIV co-infection, presence of any type of co-morbidity, having diabetes, any type of clinical complication, previous TB history, previous second-line anti-TB treatment, smear-positive at the baseline, and having XDR-TB.

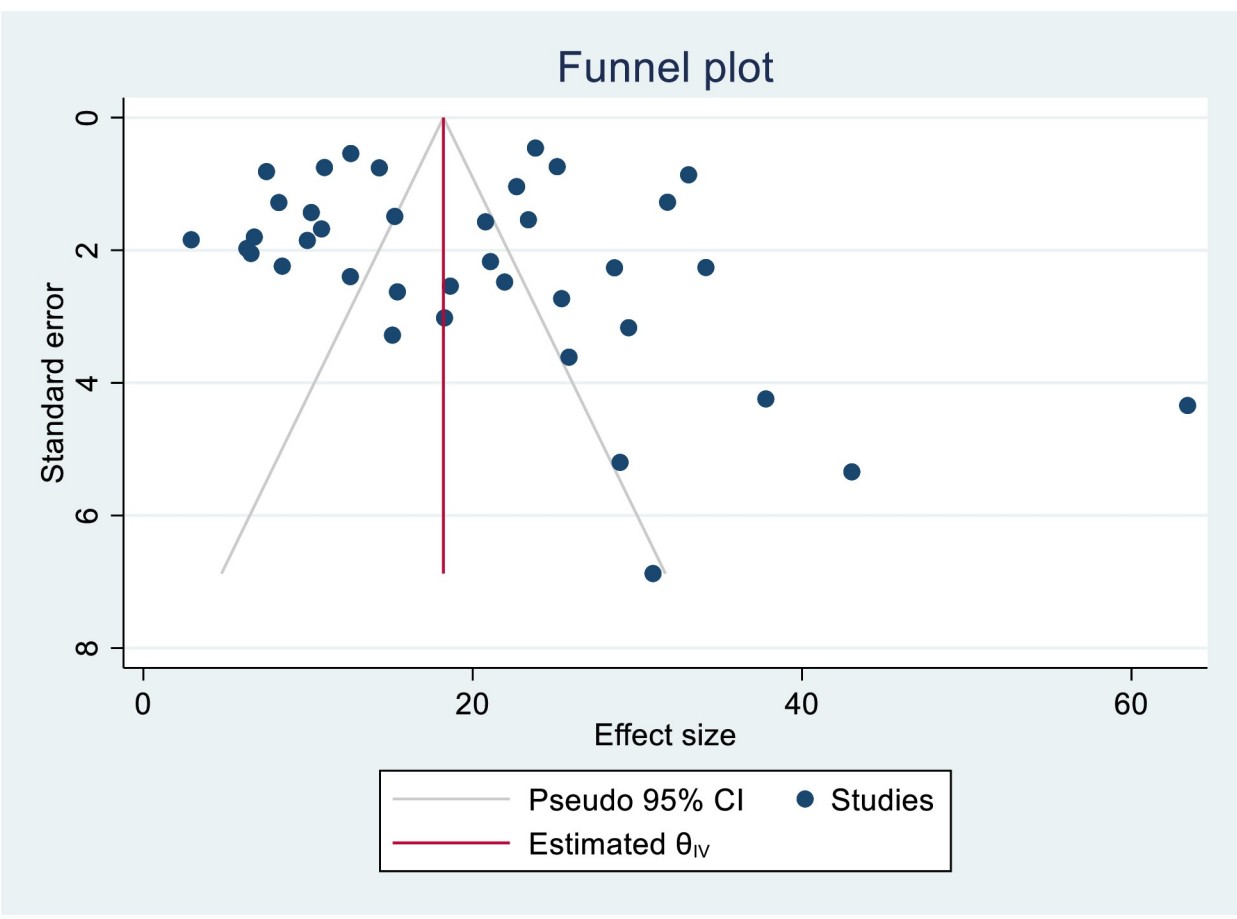

**Fig 8. Funnel plot showing publication bias among studies used to compute the incidence of mortality in patients with multi drug-resistant tuberculosis.**

The results of this study revealed that patient demographic characteristics such as male sex and older age are important predictors of mortality. Likewise, this meta-analysis revealed that male patients are more likely to die in the early TB treatment as compared to female patients. A previous study also confirmed that TB disapropriately affects males than females [65]. This could be possibly due to different factors. Males are more likely to practice smoking and alcohol drinking that might worsen the treatment outcome [66]. Additionally, evidence suggested that men are more likely to default from TB treatment, which might result in poor treatment outcomes [67]. The other demographic factor significantly associated with mortality is the age group. Oder age is operationalized in the current study as the highest age category in the individual studies and most of the studies it is above 60 years. Our pooled analysis revealed that older individuals are at a higher risk of death. As age increases in one unit, the incidence of death increases by 1%. The impaired immune status in this age group could be one factor, and older people are more likely to have other comorbidities/chronic illnesses/ that might increase the risk of mortality [68].

Another finding of this study revealed that clinical factors or patient conditions are important predictors of mortality. Those patients with any type of comorbidities were at a higher risk of death. The hazard of death in DR-TB patients with any type of comorbidities was two times as compared to their counterparts. The risk of death in comorbid DR-TB patients was

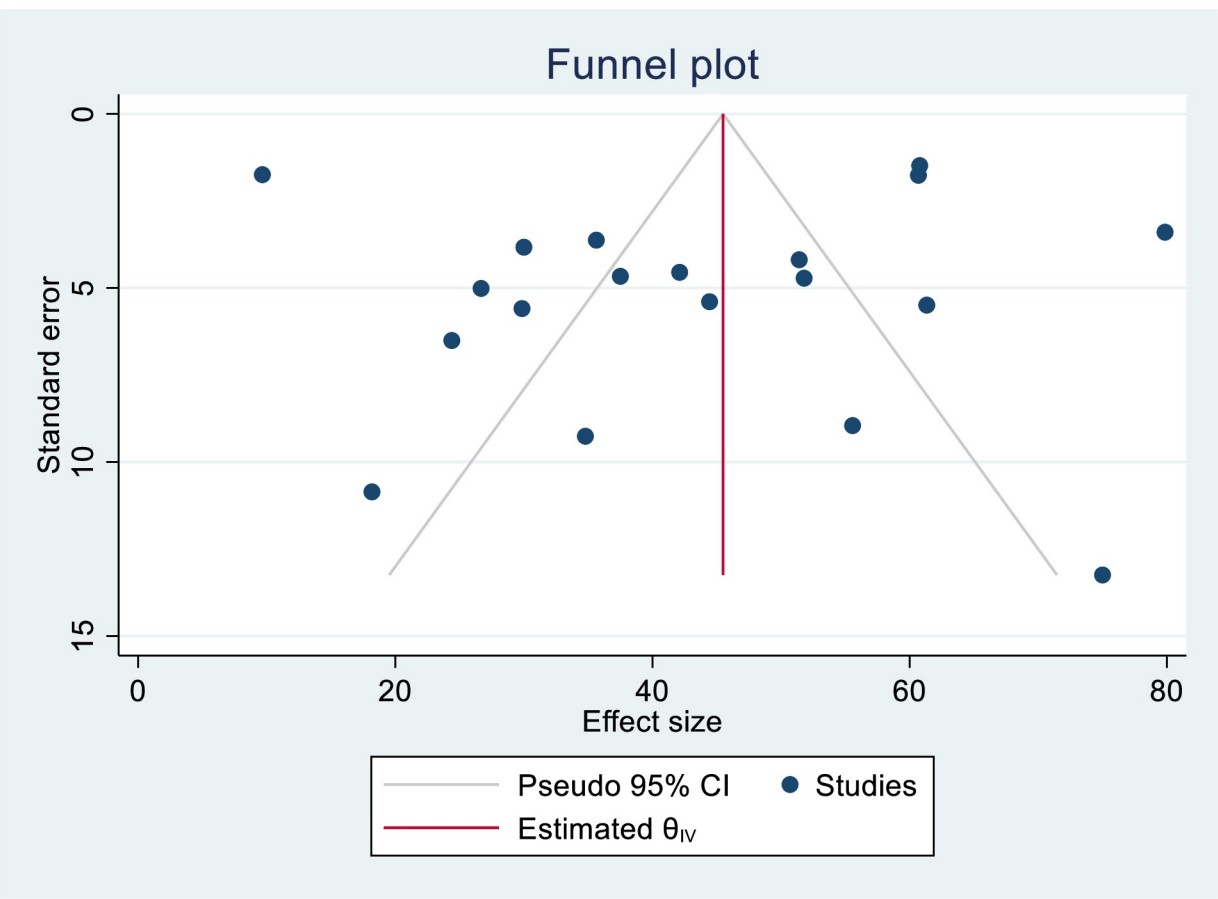

**Fig 9. Funnel plot showing publication bias among studies used to compute the incidence of mortality in patients with extensively drug-resistant tuberculosis.**

1.61 times compared with their counterparts. Among the co-morbidities, we generated a pooled estimate of DM and HIV co-infections. The risk of death increased by 92% among *HIV*-positive patients with DR-TB. Also, the risk of death in DM co-infected DR-TB patients was increased by 74%. Collaborative efforts are needed to decrease the impact of this synergy. Besides, the presence of any type of clinical complication is associated with mortality. The prognosis of DR-TB patients who developed clinical complications in their follow-up period was poor. Along with these predicting factors, undernutrition (BMI<18.5 kg/m$^2$) is one predicting factor. The risk of death in undernourished DR-TB patients was 3.13 times. Undernutrition is associated with drug toxicity that can contribute to default and could finally result in death, as described in previous studies [69–72].

The result of the current study also revealed that patients with smear-positive DR-TB at the baseline had a higher risk of death. The hazard of death among patients with smear-positive DR-TB at the baseline was 1.45 times higher as compared to smear-negative DR-TB patients. Smear-positive patients have a higher bacterial load in their sputum that reflects the infectiousness and severity of the diseases that might be associated with mortality. Also, history of TB infection and history of treatment with second-line anti-TB drugs increased the risk of death. Besides, the risk of death doubled in patients with XDR-TB. This could be due to the toxic effects of the drugs [7].

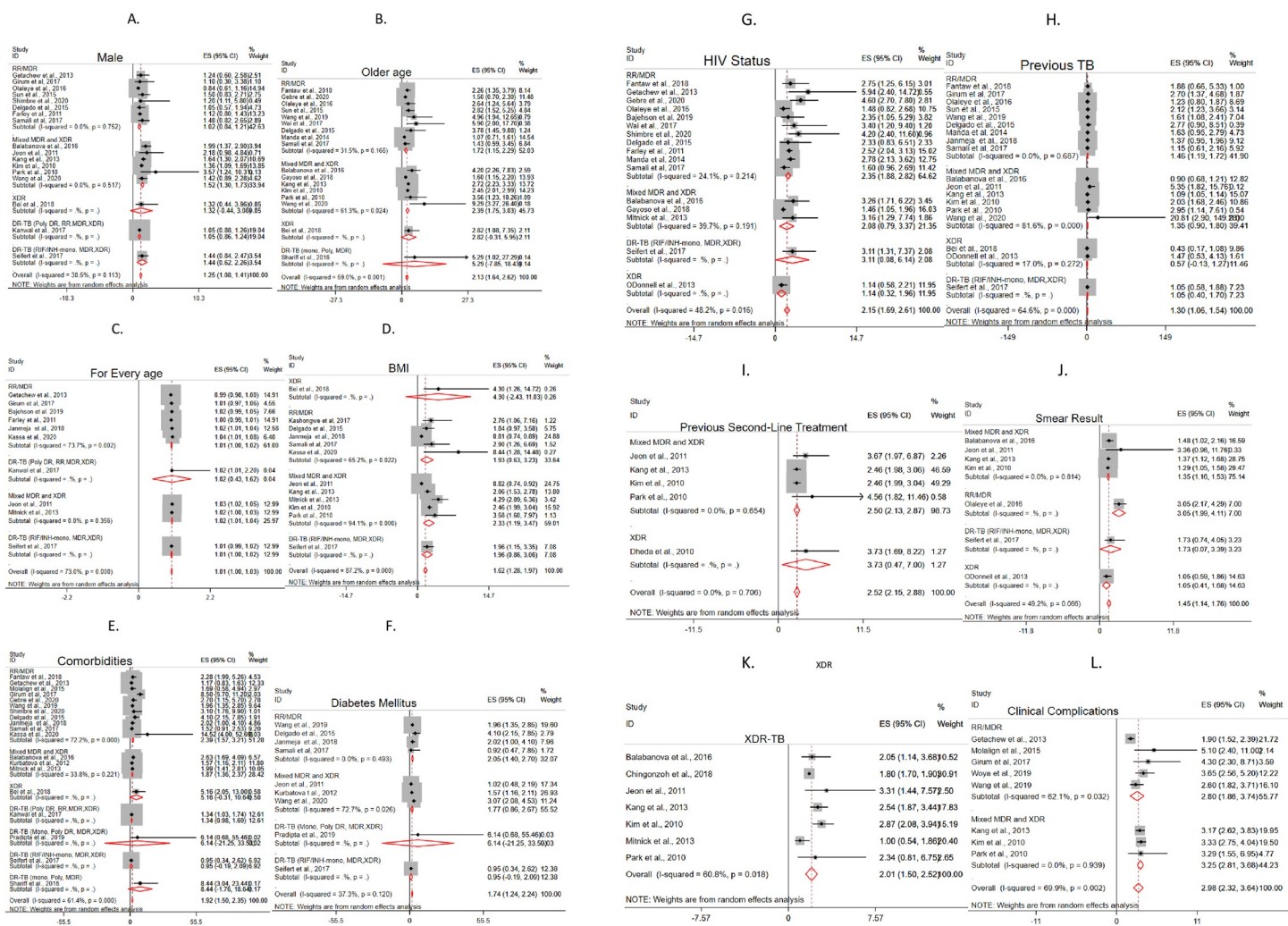

**Fig 10. Forest plot for predictors of mortality in patients with drug-resistant tuberculosis.** A. Male sex B. Older age C. For every age D. Undernutrition E. Presence of any co-morbidity F. Diabetes G. HIV co-infection H. TB history I. Previous second-line treatment J. Smear positive K. XDR-TB L. Presence of clinical complication.

In the current study, we also performed a sub-group analysis to assess the predictors of mortality across different resistance categories; 1) RR/MDR 2) mixed MDR and XDR 3) XDR 4) DR-TB. Among RR/MDR-TB patients, older age, a one-year increase in age, presence of any type of co-morbidity, having diabetes, HIV co-infection, previous TB history, being smear-positive at the baseline, and any type of clinical complications were estimated to be predictors of mortality. In the second category that is on studies that reported mortality among mixed MDR and XDR cases, the predictors of mortality include being male, older age, one year increase in age, undernutrition, presence of any type of co-morbidity, previous second-line anti-TB treatment, and smear-positive at the baseline. The predictors of mortality among XDR-TB patients include; older age, undernutrition, presence of any type of co-morbidity, and previous second-line anti-TB treatment. The fourth category is among different combinations of DR-TB including mono-resistance, poly-resistance, MDR and XDR-TB, older age, a one-year increase in age, undernutrition, and presence of any co-morbidity. Some predictors of mortality are specific to a certain group. For example, DM co-infection, HIV co-infection,

and clinical complication were found to be predictors of mortality in the first group (RR/MDR-TB) and male sex is a predictor of mortality only in the second category (mixed MDR-TB and XDR-TB). Also, being smear-positive at the baseline, and a one-year increase in age were the predictors of mortality in the first (RR/MDR-TB), and second (mixed MDR-TB and XDR-TB) categories. Previous treatment with second-line anti-TB treatment is a predictor in the second (mixed MDR-TB and XDR-TB) and third XDR-TB) categories. Thus, considering the risk factors of mortality to each drug-resistance category during anti-TB treatment would help to improve the treatment outcome. Generally, the common predictors of mortality among different drug-resistance categories identified based on the pooled estimates in the current study include; older age, presence of any type of co-morbidity, and undernutrition. Therefore, the elders need special attention during DR-TB management. Also, supportive intervention such as nutritional supplementations to DR-TB patients would improve the treatment outcome. Besides, it would be important to give special attention to DR-TB patients with underlying co-morbidities to improve the treatment outcome.

In the end, in the current study, the quality assessment revealed that the majority of the studies (44 out of 46) were graded to have high quality, and only two studies were categorized under medium quality. Also, the sensitivity analysis revealed that no single study affected the pooled incidence of mortality in drug-resistant TB patients. This implies that the quality of the studies might not affect the results in the current systematic review and meta-analysis.

## Limitation of the study

Finally, this study has some limitations. First, this study was based on studies published only in the English language. Besides, the risk factors were not separately assessed based on the place where the individual studies were conducted.

## Conclusion

In conclusion, the findings of this study revealed that different patient-related factors increased early mortality in patients with DR-TB. The presence of different co-morbidities and developing clinical complications worsen the treatment outcome in addition to the gender and age differences. Special considerations and personalized treatment and follow-up of patients with other co-morbidities, the elder ones, those who develop clinical complications, and those with previous anti-TB treatments could be essential to have a good prognosis.

## Supporting information

**S1 Table. Completed PRISMA 2009 checklist.**
(DOC)

**S2 Table. Search engines.**
(DOCX)

**S3 Table. Inclusion and exclusion criteria for selection of studies.**
(DOCX)

**S4 Table. Quality assessment for the included studies in meta-analysis.**
(DOCX)

## Acknowledgments

We acknowledge the authors of the primary studies. Our acknowledgment also goes to the Ethiopian Public Health Institute and Haramaya University College of Health Science for non-financial supports including access to internet searching.

## Author Contributions

**Conceptualization:** Ayinalem Alemu.

**Data curation:** Ayinalem Alemu.

**Formal analysis:** Ayinalem Alemu, Zebenay Workneh Bitew.

**Investigation:** Ayinalem Alemu, Zebenay Workneh Bitew, Dinka Fikadu Gamtesa.

**Methodology:** Ayinalem Alemu, Zebenay Workneh Bitew, Teshager Worku.

**Software:** Ayinalem Alemu, Zebenay Workneh Bitew, Teshager Worku.

**Writing – original draft:** Ayinalem Alemu.

**Writing – review & editing:** Ayinalem Alemu, Teshager Worku, Animut Alebel.

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
