## [Decision Letter · Decision Letter 0]

9 Feb 2021

PONE-D-20-32540

Predictors of mortality in patients with drug-resistant tuberculosis: A systematic review and meta-analysis

PLOS ONE

Dear Dr. Alemu,

Thank you for submitting your manuscript to PLOS ONE. After careful consideration, we feel that it has merit but does not fully meet PLOS ONE’s publication criteria as it currently stands. Therefore, we invite you to submit a revised version of the manuscript that addresses the points raised during the review process.

Critically, only a single manuscript version should be included in the final text.

Please take extra care to address all issues relating to the methodology

Additionally, thorough proofreading of the text is necessary

We look forward to receiving your revised manuscript.

Kind regards,

Ivan Sabol

Academic Editor

PLOS ONE

Journal Requirements:

2. Please confirm that you have included all items recommended in the PRISMA checklist including details of reasons for study exclusions in the PRISMA flowchart and number of studies excluded for each reason.

Additional Editor Comments:

The mortality was associated with “older age” separately from “1 year increase in age” suggesting “older age” is a category. However “older age” is not defined?

Table 2 lists “for one year increase” at the last row of the table. While this implies age in the context of the rest of the text, it should be revised so that the table is understandable on its own.

Some sorting of the Table 1 /Figure 2 would make the data more easy to read. For example if the studies were listed in alphabetical order, date of publication or something similar allowing easier cross referencing between table and figure.

While not an expert on meta analysis, it is strange that Figure 2 has a note mentioning weights, while no weighting information is shown. Figure 3 has the same note but also the “% weight“ column.

In the combined PDF document there are 2 versions of the manuscript, one version from page 5 to page 23. And another version from page 24 to page 39 (?). Some figures are repeated (Fig 1 at page 40 and 45. More importantly there are 2 different figures labelled Figure 4 (one at page 44 and one at page 48). One version of the manuscript specifically mentions figures 1-4 but doesn’t mention figures 5-7. Which manuscript is final?

Figure 4= Figure 7 subpanels should be made more uniform

Figure 6 is difficult to read.

Typos and trivial

The manuscript should be thoroughly proofread since it contains many typographical spacing errors starting at Line 4 with authors names

The word inconvenience(s) as used at lines 94, 113 should probably be replaced by inconsistencies

Line 441 – 442 font sizes are different

Reviewers' comments:

Reviewer's Responses to Questions

**Comments to the Author**

1. Is the manuscript technically sound, and do the data support the conclusions?

Reviewer #1: Partly

Reviewer #2: Yes

Reviewer #3: Yes

2. Has the statistical analysis been performed appropriately and rigorously? 

Reviewer #1: I Don't Know

Reviewer #2: I Don't Know

Reviewer #3: Yes

3. Have the authors made all data underlying the findings in their manuscript fully available?

Reviewer #1: Yes

Reviewer #2: Yes

Reviewer #3: Yes

4. Is the manuscript presented in an intelligible fashion and written in standard English?

Reviewer #1: No

Reviewer #2: Yes

Reviewer #3: Yes

5. Review Comments to the Author

Reviewer #1: In this systematic review, authors aimed to determine the predictors of mortality in patients with drug-resistant tuberculosis.

The major limitation of this analysis is that it does not distinguish between different types of drug resistance. They analyzed all patients with „resistance to any TB drug“ which means that patients with i.e. isoniazid- or ethambutol or PZA- monoresistance are mixed with patients with MDR and even XDR TB. Given that these are very different populations of patients with very different outcomes (according to all knowledge from the literature), it does not make sense to mix them together in order to try to have some conclusions that consequentially cannot be applied for neither on of those subgroups among the hat of „drug resistance“. Accordingly, the calculated proportion of death is 25.35, but ranges from 6.74% to 79.86%, which is, most probably, a reflection of very different populations in studied cohorts. I would suggest to differentiate patients populations and focus on a subgroup (i.e. monoresistance, MDRTB or XDR TB) in order to have information that could be useful or applied in real life.

Other remarks include many typo errors and a general need to improve the language of the manuscript as well as duplication of the figures in the uploaded file.

Reviewer #2: Dear authors - I appreciate the opportunity to read your manuscript. I was asked to review the methods, so I will restrict my comments to that section. I have a few comments.

1. The databases searched are suitable. Please include the platforms (e.g. Ovid Embase or Elsevier Embase, etc)

2. I appreciate the attempt to search grey literature.

3. Thank you for reporting the search date. It's from June 2020. Consider an update to make the search more current.

4. I appreciate the inclusion of an additional file with the searches. Rather than include screenshots of the database, please include just the searches in the document. The search terms in PubMed are cut off, for example.

5. Each search would benefit with explicit use of keywords and controlled terms. For example, the common structure would be something like this:

#1 'drug resistant tuberculosis'/exp OR 'drug resistant tuberculosis':ti,ab

#2 'mortality'/exp OR mortalit*:ti,ab OR death*:ti,ab

#3 predict*:ti,ab OR indicat*:ti,ab

#4 #1 AND #2 AND #3

This is for Elsevier Embase. Each one of these search lines could be expanded further with synonyms and other variations. But this brings in several times the results of the current search reported, and I don't think they'd all be too far afield. So, I recommend broadening the searches and reporting them more accessibly in the 'additional file'.

Reviewer #3: The reviewer congratulates the authors on this submission. Below are some comments for consideration.

Existing literature:

- There are a number of previous systematic reviews looking at outcomes of drug-resistant TB, one even written by the same team as the current review (Alemu et al 2020, Poor treatment outcome and its predictors among drug-resistant tuberculosis patients in Ethiopia: A systematic review and meta-analysis.) It would be important to emphasize the value added by the new work - what is different about the current review? (If it is just that the other review focused on treatment outcomes generally, and this one is focusing on mortality, then this should be stated).

Introduction:

- Introduction is somewhat brief, could expand upon specific prior work and findings regarding risk factors for mortality among DR TB patients, and why previous systematic reviews have been insufficient / how your review adds value.

Methods:

- Specific table of inclusion/exclusion criteria would be helpful (e.g. age range, locations of studies, etc)

- Clarify PICOS elements, e.g. for DR TB, state diagnostic criteria that participants need to fulfill in order to be included (case definition is mentioned in discussion, but should be moved up to be mentioned earlier, in methods). For outcome (mortality), state whether this was defined as death from any cause during the treatment period, or whether it was DR TB specific mortality

- Justification of using both fixed and random effects MA? Why not use random effects for all analyses given that the studies are done in different populations?

Results:

- Quality assessment results not described in results section, need to add this. In addition to description, would be good to add a visual representation, e.g. summarizing the information in Supplementary file 3.

Minor comments:

- line 133: start by mentioning total number of records returned.

Discussion:

- Need to add discussion of study quality and how this may have affected results.

Tables and Figues:

- Table 1: Not sure if this is ordered in any way but if not, could order by year, country, or author's last name.

- Figure 1: Recommend to use original PRISMA version of screening flowchart. For studies excluded at full text review, reasons for exclusion need to be stated (e.g. case reports, n=X, etc). (It seems this has now been included in revised version).

- Figure 6 - not possible to see study author names/years, need higher quality image.

- Figure 7 estimates difficult to see, suggest larger/higher quality images.

- Additional file 2: please display specific search terms in a table rather than as a screenshot of databases. The search terms are sometimes cut-off in the screenshots, so not clear to see.

General comments:

- It is mentioned that all data is contained in the manuscript and supplementary files - will code be shared as well?

- in line 31, replace "electronic five major" with "five major electronic"

- other minor grammar errors throughout, please verify.

- line 113: inconvenience? unclear. Maybe meant to say inconsistencies / discrepancies between reviewers?

6. PLOS authors have the option to publish the peer review history of their article (what does this mean?). If published, this will include your full peer review and any attached files.

Reviewer #1: No

Reviewer #2: **Yes: **Mark MacEachern

Reviewer #3: No

---

## [Author Response · Author response to Decision Letter 0]

27 Feb 2021

Revisions based on the Editor’s and the reviewers’ comments and suggestions

Title: Predictors of mortality in patients with drug-resistant tuberculosis: A systematic review and meta-analysis

Editor Comments and suggestions

1. Critically, only a single manuscript version should be included in the final text.

Thank you for the valuable comment. The problem occurred due to the second manuscript submitted after the technical check comments without removing the first submission. The current revised manuscript is a single version.

2. Please take extra care to address all issues relating to the methodology

Thank you for the comment. We revised the manuscript as per the comments and suggestions that were given by the editor and the reviewers.

3. Additionally, thorough proofreading of the text is necessary

Thank you for the constructive comment. The current manuscript is revised, and proofread by someone with better English language skills.

Additional Editor Comments:

1. The mortality was associated with “older age” separately from “1 year increase in age” suggesting “older age” is a category. However “older age” is not defined?

Thank you for the valuable comment. We operationalized “older age” in the revised manuscript as; the highest age category in the individual studies and most of the studies, it is above 60 years.

2. Table 2 lists “for one year increase” at the last row of the table. While this implies age in the context of the rest of the text, it should be revised so that the table is understandable on its own.

Thank you for the comment, now revised accordingly as a one-year increase in age.

3. Some sorting of the Table 1 /Figure 2 would make the data more easy to read. For example if the studies were listed in alphabetical order, date of publication or something similar allowing easier cross referencing between table and figure.

Thank you for the comment and suggestion. In the revised manuscript, the studies in Table 1 are listed in alphabetical order. 

4. While not an expert on meta analysis, it is strange that Figure 2 has a note mentioning weights, while no weighting information is shown. Figure 3 has the same note but also the “% weight“ column.

Thank you for the valuable comment. In the revised submission, all figures have weighting information

5. In the combined PDF document there are 2 versions of the manuscript, one version from page 5 to page 23. And another version from page 24 to page 39 (?). Some figures are repeated (Fig 1 at page 40 and 45. More importantly there are 2 different figures labelled Figure 4 (one at page 44 and one at page 48). One version of the manuscript specifically mentions figures 1-4 but doesn’t mention figures 5-7. Which manuscript is final?

Figure 4= Figure 7 subpanels should be made more uniform

Figure 6 is difficult to read.

Thank you for the valuable comment; this was due to the mix-up of two uploaded manuscripts. The current manuscript is revised accordingly. Since figure 6 is not readable due to a large number of studies, we removed it from the revised manuscript. However, we captured information. The number of figures in the revised manuscript is revised due to the suggestion of Reviewer 1 to perform a separate analysis based on the drug-resistance pattern. 

6. Typos and trivial

Thank you for the valuable comment. The entire manuscript is checked for typos and proofread by someone with better English language skills. 

7. The manuscript should be thoroughly proofread since it contains many typographical spacing errors starting at Line 4 with authors names

Thank you for the comment. It is corrected.

8. The word inconvenience(s) as used at lines 94, 113 should probably be replaced by inconsistencies

Thank you for the suggestion. We replaced it.

9. Line 441 – 442 font sizes are different.

Thank you for the comment. We corrected it. 

Reviewer 1

Reviewer #1: In this systematic review, authors aimed to determine the predictors of mortality in patients with drug-resistant tuberculosis.

1. The major limitation of this analysis is that it does not distinguish between different types of drug resistance. They analyzed all patients with „resistance to any TB drug“ which means that patients with i.e. isoniazid- or ethambutol or PZA- monoresistance are mixed with patients with MDR and even XDR TB. Given that these are very different populations of patients with very different outcomes (according to all knowledge from the literature), it does not make sense to mix them together in order to try to have some conclusions that consequentially cannot be applied for neither on of those subgroups among the hat of „drug resistance“. Accordingly, the calculated proportion of death is 25.35, but ranges from 6.74% to 79.86%, which is, most probably, a reflection of very different populations in studied cohorts. I would suggest to differentiate patients populations and focus on a subgroup (i.e. monoresistance, MDRTB or XDR TB) in order to have information that could be useful or applied in real life.

Thank you for your valuable comment and suggestion. In the revised manuscript, we analyzed the pooled estimate of mortality for MDR-TB and XDR-TB separately.

2. Other remarks include many typo errors and a general need to improve the language of the manuscript as well as duplication of the figures in the uploaded file.

Thank you for the pertinent comment. The current version is checked by someone with better English language skills, and the duplicate figures are removed. The duplicates were due to uploading the revised manuscript after a technical check without removing the first manuscript. 

Reviewer 2

Reviewer #2: Dear authors - I appreciate the opportunity to read your manuscript. I was asked to review the methods, so I will restrict my comments to that section. I have a few comments.

1. The databases searched are suitable. Please include the platforms (e.g. Ovid Embase or Elsevier Embase, etc)

Thank you for the valuable comment. We included the Ovid Embase platform in the revised manuscript. 

The search string applied for the Ovid Embase database was (‘predictors’/exp OR predictors OR ‘indicators’/exp OR indicators) AND (‘mortality’/exp OR mortality) AND ‘drug resistant’ AND (tuberculosis’/exp OR tuberculosis).

2. I appreciate the attempt to search grey literature.

3. Thank you for reporting the search date. It's from June 2020. Consider an update to make the search more current.

Thank you for the valuable comment. We tried to find newly published articles but we could not find any relevant articles for the current study. 

4. I appreciate the inclusion of an additional file with the searches. Rather than include screenshots of the database, please include just the searches in the document. The search terms in PubMed are cut off, for example.

Thank you for the comment. We uploaded the full search text of the PubMed search and included it in the additional file 2.

5. Each search would benefit with explicit use of keywords and controlled terms. For example, the common structure would be something like this:

#1 'drug resistant tuberculosis'/exp OR 'drug resistant tuberculosis':ti,ab

#2 'mortality'/exp OR mortalit*:ti,ab OR death*:ti,ab

#3 predict*:ti,ab OR indicat*:ti,ab

#4 #1 AND #2 AND #3

Thank you for the valuable comment. We included the keywords and controlled terms in the searching strategy part of the methods section.

6. This is for Elsevier Embase. Each one of these search lines could be expanded further with synonyms and other variations. But this brings in several times the results of the current search reported, and I don't think they'd all be too far afield. So, I recommend broadening the searches and reporting them more accessibly in the 'additional file'.

Thank you for the valuable comment. The search string we used for the Elsevier Embase was (Predictors OR indicators AND mortality AND drug-resistant AND tuberculosis). We tried to broaden the searches based on your recommendation, but we could not get additional relevant articles for the current study. 

Reviewer 3

Reviewer #3: The reviewer congratulates the authors on this submission. Below are some comments for consideration.

1. Existing literature:

There are a number of previous systematic reviews looking at outcomes of drug-resistant TB, one even written by the same team as the current review (Alemu et al 2020, Poor treatment outcome and its predictors among drug-resistant tuberculosis patients in Ethiopia: A systematic review and meta-analysis.) It would be important to emphasize the value added by the new work - what is different about the current review? (If it is just that the other review focused on treatment outcomes generally, and this one is focusing on mortality, then this should be stated).

Thank you for the valuable comment. We included the justification in the current manuscript as per the suggestion. 

2. Introduction:

- Introduction is somewhat brief, could expand upon specific prior work and findings regarding risk factors for mortality among DR TB patients, and why previous systematic reviews have been insufficient / how your review adds value.

Thank you for the comment. We revised it as follows. Even though there are previously conducted systematic reviews regarding the poor treatment outcome of DR-TB and its predictors, most of the studies are geographically restricted. However, there is limited information that specifically addressed the predictors of mortality among DR-TB patients at the global level. 

3. Methods:

- Specific table of inclusion/exclusion criteria would be helpful (e.g. age range, locations of studies, etc)

Thank you for the comment and suggestion. The details of the individual studies (study country, setting, period, age-group, study design, etc) included in the study are described in Table 1. Figure 1 addressed the inclusion and exclusion criteria. In this study, there is no place and time restriction with the limitation of including only those studies published in English

- Clarify PICOS elements, e.g. for DR TB, state diagnostic criteria that participants need to fulfill in order to be included (case definition is mentioned in discussion, but should be moved up to be mentioned earlier, in methods). For outcome (mortality), state whether this was defined as death from any cause during the treatment period, or whether it was DR TB specific mortality.

Thank you for the pertinent comment. We revised it and the diagnostic criteria we used is included. “Drug-resistant tuberculosis, defined when someone is infected with Mycobacterium tuberculosis, which is resistant to at least one first-line anti-TB drug…already found in the methods part”. “The laboratory diagnostic methods to rule out DR-TB could be conventional phenotypic drug-susceptibility test or molecular methods like Xpert MTB/RIF assay and Line Probe Assay (MTBDRplus, MTBDRsl)”. 

- Justification of using both fixed and random effects MA? Why not use random effects for all analyses given that the studies are done in different populations?

Thank you for the valid comment. We planned to perform a fixed or random effect model based on the heterogeneity, unfortunately, since the studies are from different populations all the pooled estimates were based on the random effect model. 

4. Results:

- Quality assessment results not described in results section, need to add this. In addition to description, would be good to add a visual representation, e.g. summarizing the information in Supplementary file 3.

Thank you for the comment. We revised as per the suggestion. Based on the results found through the JBI quality assessment tool, the indicators were turned in to 100% and graded as high if >80%, medium between 60–80%, and low <60%. Accordingly, the majority of the studies (44 out of 46) graded to have high quality, and only two studies were categorized under medium quality.

5. Minor comments:

- line 133: start by mentioning total number of records returned.

Thank you, we revised it. 

“From the whole search, we assessed 640 articles for eligibility. After 203 studies were removed by duplication, 437 articles were screened by title and abstract. Then, 351 articles were excluded and full-text screening was conducted on 86. Accordingly, 41 studies were excluded from the study due to mixed study groups (12), overlapped studies (10), did not have the specific outcome (9) and incomplete records (9)”. 

6. Discussion:

- Need to add discussion of study quality and how this may have affected results.

Thank you for the valuable comment; we included it in the revised manuscript.

7. Tables and Figues:

- Table 1: Not sure if this is ordered in any way but if not, could order by year, country, or author's last name.

Thank you for the valuable comment and suggestion, now it is alphabetically ordered using the Authors name.

- Figure 1: Recommend to use original PRISMA version of screening flowchart. For studies excluded at full text review, reasons for exclusion need to be stated (e.g. case reports, n=X, etc). (It seems this has now been included in revised version).

Thank you for the comment. We already corrected it during the technical check.

- Figure 6 - not possible to see study author names/years, need higher quality image.

Thank you for the comment. Due to a large number of studies, it is not visible, thus we removed it from the revised manuscript. However, we capture the information. 

- Figure 7 estimates difficult to see, suggest larger/higher quality images.

 Thank you for the comment and suggestion. It is due to the small size figures, but its quality is good. If this manuscript is accepted the size would be decided during production. 

- Additional file 2: please display specific search terms in a table rather than as a screenshot of databases. The search terms are sometimes cut-off in the screenshots, so not clear to see.

 Thank you for the valuable comment; we revised it.

General comments:

- It is mentioned that all data is contained in the manuscript and supplementary files - will code be shared as well? 

Thank you for the question, yes it is possible.

- in line 31, replace "electronic five major" with "five major electronic"

Thank you; now corrected.

- other minor grammar errors throughout, please verify.

- line 113: inconvenience? unclear. Maybe meant to say inconsistencies / discrepancies between reviewers?

Thank you for the question. Now corrected to inconsistencies.

---

## [Decision Letter · Decision Letter 1]

19 Apr 2021

PONE-D-20-32540R1

Predictors of mortality in patients with drug-resistant tuberculosis: A systematic review and meta-analysis

PLOS ONE

Dear Dr. Alemu,

Thank you for submitting your manuscript to PLOS ONE. After careful consideration, we feel that it has merit but does not fully meet PLOS ONE’s publication criteria as it currently stands.

The critical concern of reviewer 1 remains unmet after revision. Thus another reviewer was invited to additionally assess the study and raised similar concerns. The data was inappropriately combined and thus the conclusions drawn fail the publication criteria 4. https://journals.plos.org/plosone/s/criteria-for-publication

Therefore, we invite you to submit a revised version of the manuscript that addresses the points raised during the review process and critically address the issue of MDR and XDR data in your predictors of mortality analysis.

We look forward to receiving your revised manuscript.

Kind regards,

Ivan Sabol

Academic Editor

PLOS ONE

Reviewers' comments:

Reviewer's Responses to Questions

**Comments to the Author**

1. If the authors have adequately addressed your comments raised in a previous round of review and you feel that this manuscript is now acceptable for publication, you may indicate that here to bypass the “Comments to the Author” section, enter your conflict of interest statement in the “Confidential to Editor” section, and submit your "Accept" recommendation.

Reviewer #1: (No Response)

Reviewer #3: (No Response)

Reviewer #4: (No Response)

2. Is the manuscript technically sound, and do the data support the conclusions?

Reviewer #1: Partly

Reviewer #3: Yes

Reviewer #4: Partly

3. Has the statistical analysis been performed appropriately and rigorously? 

Reviewer #1: Yes

Reviewer #3: Yes

Reviewer #4: Yes

4. Have the authors made all data underlying the findings in their manuscript fully available?

Reviewer #1: Yes

Reviewer #3: Yes

Reviewer #4: (No Response)

5. Is the manuscript presented in an intelligible fashion and written in standard English?

Reviewer #1: Yes

Reviewer #3: Yes

Reviewer #4: Yes

6. Review Comments to the Author

Reviewer #1: Authors have partly addressed major remarks and have done the analyse of the pooled estimate of mortality for MDR-TB and XDR-TB separately. Still, the risk factors were assessed for all DR-TB as a group. Given that these are very different populations of patients with very different outcomes (according to all knowledge from the literature), it does not make sense to mix them together as the conclusions that consequentially cannot be applied for neither on of those subgroups among the hat of „drug resistance“.

Given the amount of quality work and analysis that was put into this manuscript, it would be of a great value to do the risk analyse separately for different groups of resistance (or at least mono-resistance vs MDR and XDR) as it would have much more sense and value for every day clinical work. I would suggest at least to put a subgroup analysis in a supplemental (if the numbers of subgroups under-power the analysis)

Reviewer #3: The reviewer thanks the authors for their efforts to address reviewer suggestions. Some minor comments remain unaddressed, see below:

1. Introduction:

Introduction still somewhat brief - need to discuss findings of prior reviews. Nice to see you have now mentioned that other reviews exist on the topic, but you have not cited them. Suggest to cite them where you first mention them (Line 77 of the introduction), and briefly mention their findings and what remains to be explored.

2. Methods:

As mentioned in the previous round of reviews, a table clearly listing the inclusion and exclusion criteria would be helpful. Although you have mentioned reasons for exclusion in the Figure 1 flowchart, these are vague (e.g. “mixed study groups”, “overlapped studies”). It would be good to have a table of minimum requirements for inclusion, i.e. which study designs included, minimum requirements for data reported to be included, etc…

3. Additional file 2:

Please display specific search terms in a table rather than as a screenshot of databases. The author response sheet says this has been corrected but I am not sure where, as the screenshots are still in additional file 2.

Reviewer #4: I have read with interest the article by Alemu and co-authors. This systematic review and meta-analysis, aiming to identify predictors of mortality for patients with multidrug-resistant tuberculosis, has already been reviewed in detail by three Reviewers. I will therefore limit my comments to high-level overall considerations.

1. In agreement with Reviewer 1, I think that the main criticism to this study is related to the inclusion of all types of drug-resistance. Pooling resistance “to at least one first-line anti-TB drug” does not make sense from a clinical perspective. I appreciate that the authors tried, in the revised version of the manuscript, to analyse separately mortality rates for MDR-TB and XDR-TB. However, this is not applied to the main objective of the study, the identification of predictors of mortality. And, this does not solve the issue of including patients with monoresistance to a TB drug (like isoniazid-monoresistance for instance) which have different mortality risk and may have different mortality predictors. My advice, as suggested by Reviewer 1, would be to restrict the systematic review to specific groups of drug resistance (for example, a) isoniazid-resistant, rifampicin-susceptible; b) rifampicin-resistant, fluoroquinolone-susceptible; and c) rifampicin- and fluoroquinolone-resistant) and identify mortality rates and predictors for ach of this groups. This way, the results would be clear and more meaningful.

2. Performing and analysis stratified by clinically-relevant subgroups of drug-resistant TB patients, as suggested in point 1, would also increase the novelty and interest of the results of this study, compared to existing literature (as per comment of Reviewer 3).

3. The search should be updated to identify more recent studies, as noted by Reviewer 2.

7. PLOS authors have the option to publish the peer review history of their article (what does this mean?). If published, this will include your full peer review and any attached files.

Reviewer #1: No

Reviewer #3: No

Reviewer #4: No

---

## [Author Response · Author response to Decision Letter 1]

10 May 2021

Revisions based on the Editor’s and the reviewers’ comments and suggestions

Title: Predictors of mortality in patients with drug-resistant tuberculosis: A systematic review and meta-analysis

Editor Comments and suggestions

1. The critical concern of reviewer 1 remains unmet after revision. Thus another reviewer was invited to additionally assess the study and raised similar concerns. The data was inappropriately combined and thus the conclusions drawn fail the publication criteria 4. https://journals.plos.org/plosone/s/criteria-for-publication

Therefore, we invite you to submit a revised version of the manuscript that addresses the points raised during the review process and critically address the issue of MDR and XDR data in your predictors of mortality analysis.

Thank you for the valuable comments and suggestions. We performed a sub-group analysis on the predictors of mortality per the types of resistance categories based on the data presented in the primary studies. Besides, we broadened our searching and four primary studies included in the revised manuscript. We performed the pooled analysis by considering these new studies. 

Reviewers’ comments and suggestions

Reviewer #1

1. Reviewer #1: Authors have partly addressed major remarks and have done the analyse of the pooled estimate of mortality for MDR-TB and XDR-TB separately. Still, the risk factors were assessed for all DR-TB as a group. Given that these are very different populations of patients with very different outcomes (according to all knowledge from the literature), it does not make sense to mix them together as the conclusions that consequentially cannot be applied for neither on of those subgroups among the hat of „drug resistance“.

Given the amount of quality work and analysis that was put into this manuscript, it would be of a great value to do the risk analyse separately for different groups of resistance (or at least mono-resistance vs MDR and XDR) as it would have much more sense and value for every day clinical work. I would suggest at least to put a subgroup analysis in a supplemental (if the numbers of subgroups under-power the analysis)

Thank you for the constructive comment. As per the suggestion given, we performed a sub-group analysis per the resistance category presented in the primary studies. 

Reviewer #3: 

Reviewer #3: The reviewer thanks the authors for their efforts to address reviewer suggestions. Some minor comments remain unaddressed, see below:

1. Introduction:

Introduction still somewhat brief - need to discuss findings of prior reviews. Nice to see you have now mentioned that other reviews exist on the topic, but you have not cited them. Suggest to cite them where you first mention them (Line 77 of the introduction), and briefly mention their findings and what remains to be explored.

Thank you for the comment once again. In the revised we cited some previous studies and also we presented the findings of a systematic review and meta-analysis study performed by our team in 2020 that showed the incidence of poor treatment outcome and its predictors among DR-TB patients in Ethiopia. The study also estimated the mortality incidence density rate.

2. Methods:

As mentioned in the previous round of reviews, a table clearly listing the inclusion and exclusion criteria would be helpful. Although you have mentioned reasons for exclusion in the Figure 1 flowchart, these are vague (e.g. “mixed study groups”, “overlapped studies”). It would be good to have a table of minimum requirements for inclusion, i.e. which study designs included, minimum requirements for data reported to be included, etc…

Thank you for the comment and suggestion. In the revised manuscript, we prepared a table for inclusion and exclusion criteria as a supplementary table 1. 

3. Additional file 2:

Please display specific search terms in a table rather than as a screenshot of databases. The author response sheet says this has been corrected but I am not sure where, as the screenshots are still in additional file 2.

Thank you for the valuable comment. Previously we presented the PubMed search terms in the table, but in the current version, we displayed the search terms for the remaining search engines in table. 

Reviewer #4:

Reviewer #4: I have read with interest the article by Alemu and co-authors. This systematic review and meta-analysis, aiming to identify predictors of mortality for patients with multidrug-resistant tuberculosis, has already been reviewed in detail by three Reviewers. I will therefore limit my comments to high-level overall considerations.

1. In agreement with Reviewer 1, I think that the main criticism to this study is related to the inclusion of all types of drug-resistance. Pooling resistance “to at least one first-line anti-TB drug” does not make sense from a clinical perspective. I appreciate that the authors tried, in the revised version of the manuscript, to analyse separately mortality rates for MDR-TB and XDR-TB. However, this is not applied to the main objective of the study, the identification of predictors of mortality. And, this does not solve the issue of including patients with monoresistance to a TB drug (like isoniazid-monoresistance for instance) which have different mortality risk and may have different mortality predictors. My advice, as suggested by Reviewer 1, would be to restrict the systematic review to specific groups of drug resistance (for example, a) isoniazid-resistant, rifampicin-susceptible; b) rifampicin-resistant, fluoroquinolone-susceptible; and c) rifampicin- and fluoroquinolone-resistant) and identify mortality rates and predictors for ach of this groups. This way, the results would be clear and more meaningful.

Thank you for the valuable comment and suggestion. As per the suggestion, we performed a sub-group analysis to assess the predictors of mortality among different resistance categories as presented in the primary studies. 

2. Performing and analysis stratified by clinically-relevant subgroups of drug-resistant TB patients, as suggested in point 1, would also increase the novelty and interest of the results of this study, compared to existing literature (as per comment of Reviewer 3).

Thank you for the comment. We performed analysis as per the suggestion. 

3. The search should be updated to identify more recent studies, as noted by Reviewer 2.

Thank you for the comment. We updated the search and we identified additional four studies. We analyzed the mortality rates and the predictors of mortality by including the additional identified studies.

---

## [Decision Letter · Decision Letter 2]

7 Jun 2021

PONE-D-20-32540R2

Predictors of mortality in patients with drug-resistant tuberculosis: A systematic review and meta-analysis

PLOS ONE

Dear Dr. Alemu,

Thank you for submitting your manuscript to PLOS ONE. After careful consideration, we feel that it has merit but needs a few very minor changes. Therefore, we invite you to submit a revised version of the manuscript that addresses the points raised during the review process.

All reviewers commend the addition of the subgroup analysis, but 2 reviewers note that this addition was not adequately included in the results and discussion and removed from the limitations. Please try to revise at least  the discussion text accordingly and mention this in the abstract at least briefly.

We look forward to receiving your revised manuscript.

Kind regards,

Ivan Sabol

Academic Editor

PLOS ONE

Journal Requirements:

Reviewers' comments:

Reviewer's Responses to Questions

**Comments to the Author**

1. If the authors have adequately addressed your comments raised in a previous round of review and you feel that this manuscript is now acceptable for publication, you may indicate that here to bypass the “Comments to the Author” section, enter your conflict of interest statement in the “Confidential to Editor” section, and submit your "Accept" recommendation.

Reviewer #1: All comments have been addressed

Reviewer #3: All comments have been addressed

Reviewer #4: (No Response)

2. Is the manuscript technically sound, and do the data support the conclusions?

Reviewer #1: Yes

Reviewer #3: Yes

Reviewer #4: Partly

3. Has the statistical analysis been performed appropriately and rigorously? 

Reviewer #1: Yes

Reviewer #3: N/A

Reviewer #4: Yes

4. Have the authors made all data underlying the findings in their manuscript fully available?

Reviewer #1: Yes

Reviewer #3: Yes

Reviewer #4: Yes

5. Is the manuscript presented in an intelligible fashion and written in standard English?

Reviewer #1: Yes

Reviewer #3: Yes

Reviewer #4: Yes

6. Review Comments to the Author

Reviewer #1: Authors have added the subgroup analyses. Few sentences regarding that should also be added in the discussion (i.e. to discuss the differences (or absence of) between the subgroups regarding the mortality predictors. Also, as this is no more a limitation of the study, the sentence under the line 281 (about not performing sub-group analysis) should be removed.

Reviewer #3: The reviewer thanks the authors for their efforts to address the reviewer’s suggestions. The reviewer has no further suggestions, as the previous suggestions have now been addressed. The reviewer recommends acceptance of the manuscript, provided remaining comments from the other reviewers have also been adequately addressed.

Reviewer #4: I thank the authors for submitting this revised version of the manuscript.

The authors should be commended for including a sub-group analysis stratified by drug resistance pattern (as requested) and for updating their search and increasing the number of papers included in the meta-analysis.

However, the sub-group analysis is currently only presented as a secondary paragraphin the results, not commented in the discussion, and not included at all in the abstract. In my opinion, what is currently presented as sub-group analysis should actually represent the main result of the study: as such, the results of the stratified analysis should be presented in the abstract and developed in the discussion section.

7. PLOS authors have the option to publish the peer review history of their article (what does this mean?). If published, this will include your full peer review and any attached files.

Reviewer #1: No

Reviewer #3: No

Reviewer #4: No

---

## [Author Response · Author response to Decision Letter 2]

9 Jun 2021

Response

Thank you to all the reviewers and the editor for the valuable comments and suggestions that improved the quality of the manuscript. At this stage, we included the predictors of mortality across different drug-resistance categories at the abstract section (line 45-47) and at the discussion section (line 276-298). We removed the sentence “Also, subgroup analysis based on the type of drug-resistance pattern was not performed.” as this is no more a limitation of the study.

---

## [Editor Report · Decision Letter 3]

15 Jun 2021

Predictors of mortality in patients with drug-resistant tuberculosis: A systematic review and meta-analysis

PONE-D-20-32540R3

Dear Dr. Alemu,

We’re pleased to inform you that your manuscript has been judged scientifically suitable for publication and will be formally accepted for publication once it meets all outstanding technical requirements.

Kind regards,

Ivan Sabol

Academic Editor

PLOS ONE
---

## [Editor Report · Acceptance letter]

18 Jun 2021

PONE-D-20-32540R3 

Predictors of mortality in patients with drug-resistant tuberculosis: A systematic review and meta-analysis 

Dear Dr. Alemu:

I'm pleased to inform you that your manuscript has been deemed suitable for publication in PLOS ONE. Congratulations! Your manuscript is now with our production department. 

Kind regards, 

on behalf of

Dr. Ivan Sabol 

Academic Editor

PLOS ONE